# Covalent organic framework membranes through a mixed-dimensional assembly for molecular separations

Hao Yang [1,2], Leixin Yang[1,2], Hongjian Wang[1,2], Ziang Xu[1,2], Yumeng Zhao[1,2], Yi Luo[1,2], Nayab Nasir[1,2], Yimeng Song[1,2], Hong Wu[1,2,3], Fusheng Pan[1,2] & Zhongyi Jiang[1,2]

Covalent organic frameworks (COFs) hold great promise in molecular separations owing to their robust, ordered and tunable porous network structures. Currently, the pore size of COFs is usually much larger than most small molecules. Meanwhile, the weak interlamellar interaction between COF nanosheets impedes the preparation of defect-free membranes. Herein, we report a series of COF membranes through a mixed-dimensional assembly of 2D COF nanosheets and 1D cellulose nanofibers (CNFs). The pore size of 0.45–1.0 nm is acquired from the sheltering effect of CNFs, rendering membranes precise molecular sieving ability, besides the multiple interactions between COFs and CNFs elevate membrane stability. Accordingly, the membranes exhibit a flux of $8.53\,kg\,m^{-2}\,h^{-1}$ with a separation factor of 3876 for n-butanol dehydration, and high permeance of $42.8\,L\,m^{-2}\,h^{-1}\,bar^{-1}$ with a rejection of 96.8% for $Na_2SO_4$ removal. Our mixed-dimensional design may inspire the fabrication and application of COF membranes.

[1] Key Laboratory for Green Chemical Technology of Ministry of Education, School of Chemical Engineering and Technology, Tianjin University, Tianjin 300072, China. [2] Collaborative Innovation Center of Chemical Science and Engineering (Tianjin), Tianjin 300072, China. [3] Tianjin Key Laboratory of Membrane Science and Desalination Technology, Tianjin University, Tianjin 300072, China. Correspondence and requests for materials should be addressed to H.W. (email: wuhong@tju.edu.cn) or to F.P. (email: fspan@tju.edu.cn) or to Z.J. (email: zhyjiang@tju.edu.cn)

Two-dimensional (2D) materials are promising molecular building blocks for separation membranes owing to their atomic thickness with minimized transport resistance as well as extraordinary physicochemical properties[1,2]. The 2D materials-based separation membranes hold great promise in energy and environment-relevant applications such as carbon capture[3], bio-alcohol fuel production[4], water purification and desalination[5]. As a kind of crystalline porous materials, 2D covalent organic frameworks (COFs) from the atomically precise integration of organic units have drawn tremendous attention in recent years[6,7]. The well-defined pore aperture, ordered one-dimensional (1D) nanochannels, readily tailored functionalities together with superior chemical stability, make 2D COFs excellent candidates for constructing new-generation membranes[8,9]. Compared with graphene oxide (GO) membranes, which are primarily prepared by filtration of GO laminates and separate targeted molecules primarily relying on the interlayer spacing[10,11], the 2D COF membranes could achieve ultrafast and highly selective separation by their abundant and well-organized in-plane nanopores as well as rigid skeletons[12]. Moreover, nanoporous COFs could offer more available monomers and assembling methods due to their structural diversities compared with the impermeable GO. However, at present there are only a few reports pertaining to 2D COF membranes for separation of small molecules[13,14]. One major reason may arise from the inferior sieving effects because the pore size (0.8–4.7 nm)[15] of chemically stable COFs (such as Schiff-base-type[16] and triazine-type COFs[17]) is larger than the kinetic diameter of most small molecules like water (0.26 nm), $CO_2$ (0.33 nm) and C1-C4 alcohols (0.38–0.51 nm). The accurate tuning of sub-nanometer pore size in COFs proves difficult by either monomer design or group modification[7,18]. Besides, fabricating COF membranes by assembling 2D COF nanosheets remains a great challenge. Particularly, the weak interlamellar $\pi$-$\pi$ interactions between assembled COF nanosheets usually result in the poor mechanical strength of the COF membranes[8,14]. It is imperative to explore an effective approach to simultaneously implement the effective pore size adjustment and mechanical strength enhancement of COF membranes.

The embedment of suitable components into the interlayers of COF nanosheets can be an efficient way to manipulate their stacking behaviors[14], and consequently the structures and properties of COF membranes. Recently, the emergence of mixed-dimensional heterostructures (2D + nD, where $n$ is 0, 1 or 3) has opened avenues for fundamental scientific topics and applied device designs[19], which inspires us to design COF membranes with high performance by a mixed-dimensional strategy. Among them, the 2D + 1D design is expected an effective approach to adjust the pore size of COF membranes. 1D materials with highly anisotropic shape are considered as potential candidates to decrease the size of pore entrance by covering onto the surface of COF nanosheets (sheltering effect) without the significant sacrifice of permeability of COF membranes. To enhance the mechanical strength of COF membranes, the 1D components should possess abundant active sites and generate robust linkage with COFs through multiple interactions.

Herein, we propose a kind of COF membranes using 2D COF nanosheets and 1D cellulose nanofibers (CNFs) as building blocks. CNFs, with 1D shape derived from plants, are one of the strongest natural materials owing to their hydrogen-bonded parallel chains[20,21], and the abundant hydroxyl groups on their surface can be conveniently utilized for subsequent functionalization. The mixed-dimensional nanocomposites comprising COF nanosheets covered by CNFs are first prepared and then assembled into the densely interlocked COF membranes in one-step. The sheltering effect of CNFs reduces the size of pore entrance of COF nanosheets and also establishes robust interlamellar microporous networks. The interlamellar equivalent micropore size in membranes can be tailored and the molecular sieving has been intensified, where smaller molecules pass through while larger ones are rejected[22]. Furthermore, the multiple interlamellar interactions render the COF membranes high mechanical strength. The resulting membranes exhibit superior performance in solvent dehydration, dye rejection and salt rejection. The mixed-dimensional design in our study is broadly applicable, which provides a general toolbox for designing robust COF membranes with high performance.

## Results

**Mixed-dimensional assembly of 2D COFs and 1D CNFs**. A Schiff-base-type COF $TpTG_{Cl}$ was synthesized by 1,3,5-tri-formylphloroglucinol (Tp) and triaminoguanidinium chloride ($TG_{Cl}$) and used as the 2D component[23,24]. The COF $TpTG_{Cl}$ can be easily exfoliated to obtain chemically stable nanosheets. Meanwhile, the intrinsic positive charge of guanidinium units on $TpTG_{Cl}$ framework makes it a good candidate to fabricate robust nanocomposites by assembling with negatively charged components. TEMPO-oxidized CNFs[25] with a high density of carboxyl groups were employed as the 1D components. A schematic diagram of the assembly process of $TpTG_{Cl}$ and CNFs is displayed in Fig. 1a. The molecular structure and synthesis procedure of $TpTG_{Cl}$ are shown in Supplementary Fig. 1. The transmission electron microscopy (TEM) image (Fig. 1b) shows that the $TpTG_{Cl}$ with 1–2 μm of lateral dimension is transparent to the electron beams. The high-resolution TEM (HRTEM) image along with fast Fourier transformation (FFT) and inverse FFT (IFFT) images (Fig. 1c) indicate the high crystallinity of $TpTG_{Cl}$ and hexagonal structures of the basal planes. The element distribution mappings and scanning electron microscopy (SEM) image of $TpTG_{Cl}$ are shown in Supplementary Figs. 2 and 3. The thickness of $TpTG_{Cl}$ nanosheets is around 1.5 nm measured by atomic force microscopy (AFM) (Fig. 1d). The powder X-ray diffraction (PXRD) pattern and $^{13}C$ CP-MAS solid-state NMR spectroscopy (Supplementary Figs. 4 and 5) confirm the crystallographic and chemical structure of $TpTG_{Cl}$[23]. The permanent porosity of $TpTG_{Cl}$ was verified by $N_2$ adsorption-desorption isotherms (Supplementary Fig. 6). The measured Brunauer-Emmet-Teller (BET) surface area is 305 $m^2 g^{-1}$ and the major pore size is 1.3 nm (Supplementary Fig. 7). TEMPO-oxidized CNFs with a diameter of around 2 nm and a length of 0.5–1 μm were observed from the AFM image (Fig. 1e). The stable mixed-dimensional nanocomposites with a planar $TpTG_{Cl}$ covered with dense networks of 1D CNFs (Fig. 1f and Supplementary Fig. 8) were spontaneously formed by blending the $TpTG_{Cl}$ and CNFs in an aqueous solution. The mixed-dimensional nanocomposites are denoted as $TpTG_{Cl}$@CNFs-X, where the serial number X = 1, 2, 3, 4, 5, 6 or 7 corresponds to increasing CNF content as shown in Supplementary Table 1. The $TpTG_{Cl}$@CNFs-X nanocomposites can be easily dispersed in water to form a stable colloidal solution (Fig. 1g), which is a preferential precursor for membrane fabrication. PXRD patterns of the as-prepared $TpTG_{Cl}$@CNFs-X nanocomposites (Supplementary Fig. 9) prove that the crystallographic structures of $TpTG_{Cl}$ and CNFs are unaffected by the assembly. The negatively charged carboxyl groups on CNFs are considered to have been integrated with the positively charged guanidine groups on $TpTG_{Cl}$ via electrostatic interactions. The charge carrying properties of the $TpTG_{Cl}$ solution and CNFs solution were characterized by zeta potential measurement at different pH values to confirm the electrostatic interactions (Supplementary Fig. 10). Both the $TpTG_{Cl}$ and CNFs exhibit dominant surface charges when the pH is in the range of 2 to 6.

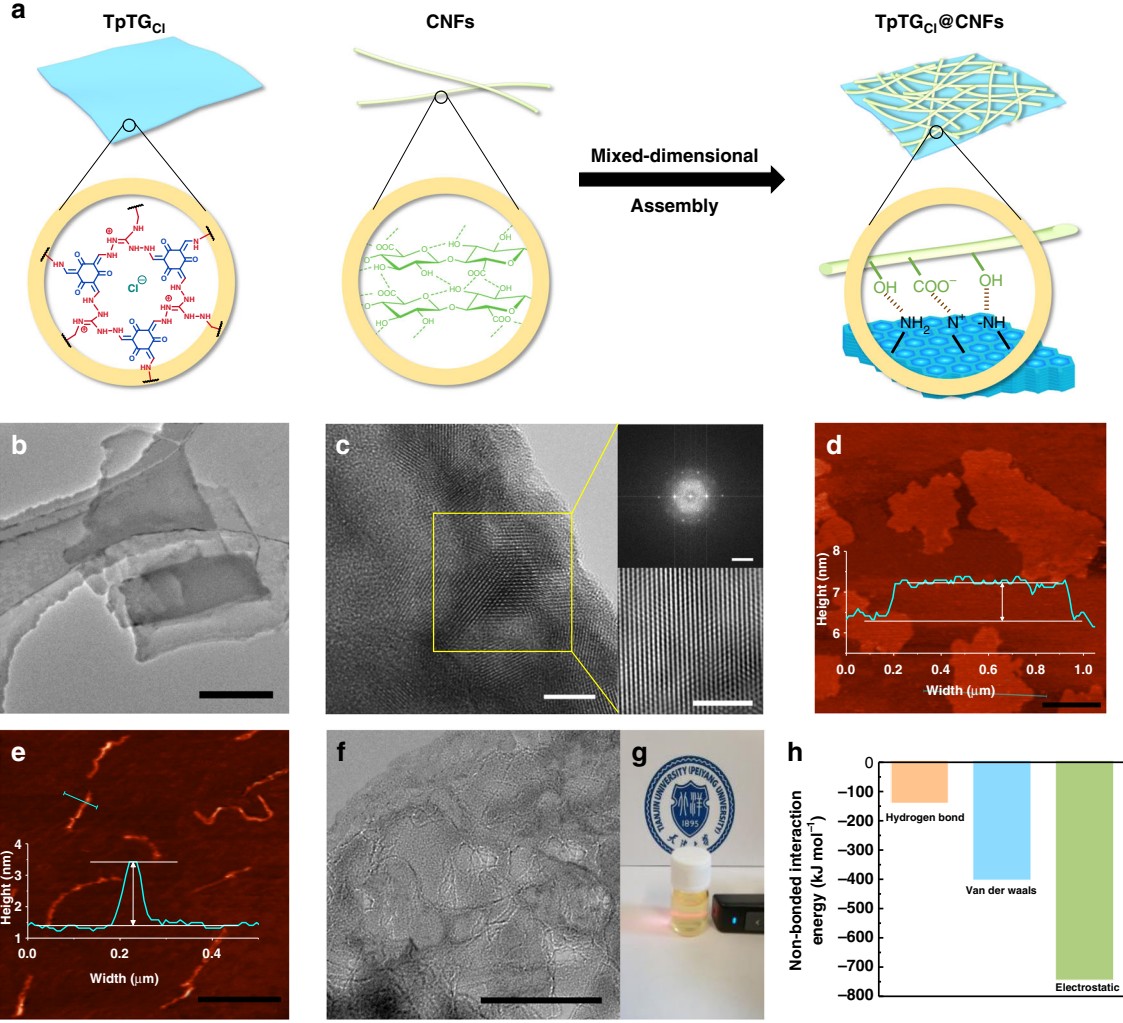

**Fig. 1** Mixed-dimensional assembly of TpTG$_{Cl}$ and CNFs. **a** A schematic illustration showing the assembly process and the interactions. **b** TEM image of TpTG$_{Cl}$ (scale bar, 500 nm). **c** HRTEM image of TpTG$_{Cl}$ (scale bar, 5 nm) with FFT (top-right, scale bar, 2 nm$^{-1}$) and IFFT images (bottom-right, scale bar, 5 nm). **d** AFM image and height profile of TpTG$_{Cl}$ (scale bar, 500 nm). **e** AFM image and height profile of CNFs (scale bar, 1 μm). **f** TEM image of TpTG$_{Cl}$@CNFs-X nanocomposites (scale bar, 100 nm). **g** Tyndall scattering effect in the aqueous solution of TpTG$_{Cl}$@CNFs-X nanocomposites. **h** MD simulations of the calculated non-bonded interaction energy between TpTG$_{Cl}$ and CNFs

Therefore, the electrostatic interactions between TpTG$_{Cl}$ and CNFs can be readily established in near-neutral solutions. Fourier transform infrared spectroscopy (FTIR) and X-ray photoelectron spectroscopy (XPS) were employed to further characterize the interactions between TpTG$_{Cl}$ and CNFs (Supplementary Figs. 11 and 12). In addition, TpTG$_{Cl}$ can form numerous hydrogen bonds with CNFs since imine groups and hydroxyl/carboxyl groups can both act as donor or acceptor of the hydrogen bonds[26]. We also performed all-atom molecular dynamics (MD) simulations to elucidate the assembly behavior. In the simulations, CNFs are placed nearby the TpTG$_{Cl}$ surface, and the non-bonded interactions are formed between TpTG$_{Cl}$ and CNFs after the structure optimization. The results (Fig. 1h) show that electrostatic interaction is the main contributor towards the non-bonded interaction energy between TpTG$_{Cl}$ and CNFs. The hydrogen bond interactions also play a key role to facilitate assembly behavior.

**Preparation of COF membranes**. The COF membrane was fabricated by a vacuum-assisted self-assembly method on a polyacrylonitrile (PAN) substrate (Fig. 2a). The PAN substrate with an average surface pore size of 30 nm (Supplementary

Fig. 13) was uniformly deposited by a dense and defect-free TpTG$_{Cl}$@CNFs-X membrane, as shown in SEM and AFM images (Fig. 2b and Supplementary Fig. 14). The as-prepared TpTG$_{Cl}$@CNFs-X membranes are flexible and their thickness can be tuned from dozens of nanometers to a few microns by varying the volume of the filtrate (Fig. 2c and Supplementary Figs. 15 and 16). The cross-section TEM image of the TpTG$_{Cl}$@CNFs-X membrane reveals an obvious lamellar structure (Fig. 2d). For obtaining a free-standing TpTG$_{Cl}$@CNFs-X membrane, the TpTG$_{Cl}$@CNFs-X membrane with a thickness of more than 2 μm was filtered on a hydrophobic poly (ether sulfone) (PES) microfiltration membrane (pore size is around 0.22 μm). After drying, the free-standing membrane could be easily peeled off from the PES substrate and used for characterization. The PXRD pattern of the TpTG$_{Cl}$@CNFs-X membrane is shown in Supplementary Fig. 17, which also displays high intensity of (100) and (001) peaks of COF TpTG$_{Cl}$.

**The pore size distribution and stability of the membranes**. A schematic diagram of pore structures in the membranes is displayed in Fig. 3a. For the TpTG$_{Cl}$@CNFs-X nanocomposites, the sheltering effect of CNFs reduces the size of pore entrance of

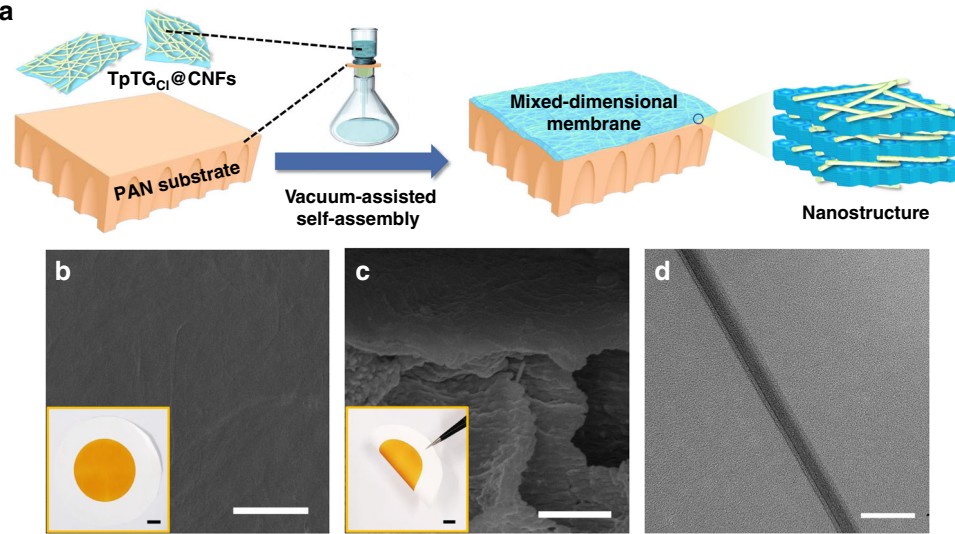

**Fig. 2** Morphology of the membranes. **a** A schematic illustration showing the vacuum-assisted self-assembly and the mixed-dimensional nanostructure. **b** Surface SEM image of the TpTG$_{Cl}$@CNFs-5/PAN membrane (scale bar, 1 μm) inserted with a digital photo of the membrane (scale bar, 1 cm). **c** Cross-sectional SEM image of the TpTG$_{Cl}$@CNFs-5/PAN membrane (scale bar, 1 μm) inserted a tweezer bent membrane photo (scale bar, 1 cm). **d** Cross-sectional TEM image of the TpTG$_{Cl}$@CNFs-5 membrane (scale bar, 50 nm)

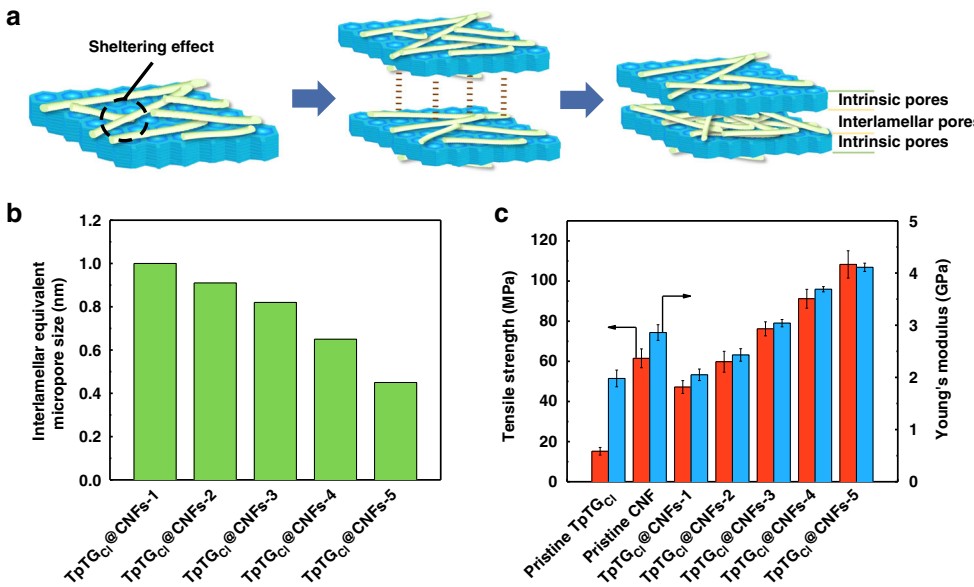

**Fig. 3** Pore size distribution and mechanical strength of the membranes. **a** A schematic diagram showing the sheltering effect, interlamellar interactions and pore structures in the membranes. **b** The interlamellar equivalent micropore size of the TpTG$_{Cl}$@CNFs-X membranes. **c** The mechanical properties of the TpTG$_{Cl}$@CNFs-X membranes. Error bars represent standard deviations for 3 measurements

TpTG$_{Cl}$ framework. After assembling the TpTG$_{Cl}$@CNFs nanocomposites into membranes, the interlamellar pores, consisting of network pores of CNFs, and pore entrance of TpTG$_{Cl}$ are formed. The pore size distributions of the membranes are demonstrated in Supplementary Fig. 18. For all membranes, the pore size distribution of 1.3 nm is derived from the intrinsic pores of TpTG$_{Cl}$. The pore size distribution of less than 1.3 nm is derived from the interlamellar micropores in membranes. With increasing CNF content in membranes, the number of interlamellar micropores increases while the interlamellar equivalent micropore size declines from 1.0 to 0.45 nm (Fig. 3b). The pore size distribution variation is mainly due to the interlocked CNF networks and the sheltering of CNFs on the nanopores of TpTG$_{Cl}$. This also manifests that the pore size distribution in the

membranes could be effectively tailored to accomplish different molecular separations.

The mechanical properties of the free-standing membranes are shown in Fig. 3c. The TpTG$_{Cl}$@CNFs-X membranes show much higher tensile strength (up to 108.3 MPa) and elastic modulus (up to 3.62 GPa) compared with the pristine TpTG$_{Cl}$ membranes and the pristine CNF membrane. The high mechanical properties of the TpTG$_{Cl}$@CNFs-X membranes are mainly attributed to the following three structural features: (i) the horizontal orientation of TpTG$_{Cl}$; (ii) hydrogen-bonded parallel chains of 1D CNFs and (iii) the multiple interactions between TpTG$_{Cl}$ and CNFs[27]. Furthermore, the structures of the membranes are similar to the bee honeycombs in nature, which consists of multilayered hexagonal cells and fiber-like cell covers and exhibits excellent

mechanical strength[28]. A fracture model is proposed to elucidate the enhancement of the mechanical strength of the membranes (Supplementary Fig. 19). For the pristine TpTG$_{Cl}$ membrane, the weak interlamellar $\pi$-$\pi$ interactions make it vulnerable when stretching. While for the TpTG$_{Cl}$@CNFs-X membrane, the hydrogen bonds of CNF networks are initially broken, and then CNFs are stretched and energy is dissipated[27]. On further stretching, the ionic bonds between TpTG$_{Cl}$ and CNFs are broken and more energy is absorbed which brings about the fracture of the membrane. Thermogravimetric analysis (TGA) curves show the high thermal stability of the membranes, in which the weight loss can be as low as 5% when the temperature is below 200 °C (Supplementary Fig. 20). The chemical and solvent stability of the membranes were verified by the fact that the membranes remained intact even after 15 d of solvent or acid treatment. The PXRD analysis was performed on these treated membranes, which confirmed that the structures remained unaltered[12] (Supplementary Fig. 21). The chemical resistance property of these membranes arises from the ketoenamine-based backbone of TpTG$_{Cl}$[16], hydrogen-bonded chains of CNFs, and the multiple interactions between TpTG$_{Cl}$ and CNFs. Thus, these membranes can find many practical separation applications.

**Water-selective permeation properties of the membranes**. In industry, solvent dehydration is a frequently-used requisite purification process and it is commonly energy-intensive and costly, especially for the (near-) azeotropic systems such as water/alcohol mixtures[29,30]. The water-selective permeation properties of the membranes were tested by a pervaporation process (Supplementary Fig. 22) using water/alcohol binary mixtures (10 wt%/90 wt%) under a strong 60 L h$^{-1}$ cross flow. The separation performance of the membranes was optimized by manipulating the CNF content and the thickness of TpTG$_{Cl}$@CNFs-X layer for different water/alcohol systems (Fig. 4a and Supplementary Fig. 23 and Supplementary Table 2). The C2-C4 alcohols were readily retained by the as-prepared TpTG$_{Cl}$@CNFs-5/PAN membrane and the water concentration in permeate was higher than 99.5 wt%, corresponding to separation factors ($\alpha_{water/alcohol}$) higher than 2000. In particular, the high separation performance with a permeation flux of 8.53 kg m$^{-2}$ h$^{-1}$ and a $\alpha_{water/n-butanol}$ of 3876 was achieved for the water/n-butanol mixture. Compared with the TpTG$_{Cl}$@CNFs-5 membrane, the membranes with fewer CNFs exhibit lower separation factor, and those with more CNFs exhibit lower permeation flux because the interlamellar equivalent micropore size in membranes decreases with the increasing CNF content. The effect of operation temperature on the separation performance of the TpTG$_{Cl}$@CNFs-5/PAN membrane was investigated, and the permeance and selectivity of the membranes were calculated (Supplementary Fig. 24). A long-term operation was carried out to assess the stability of the TpTG$_{Cl}$@CNFs-5/PAN membrane (Fig. 4b). The permeation flux was higher than 7 kg m$^{-2}$ h$^{-1}$ and the water concentration in permeate was more than 99.8 wt% after 15 d. The steady permeation flux and the high $\alpha_{water/n-butanol}$ further confirm the high stability of the TpTG$_{Cl}$@CNFs-5/PAN membrane in solvents and also demonstrate their potential in practical applications. Moreover, the TpTG$_{Cl}$@CNFs-5/PAN membrane in this study exhibits superior separation performance compared with the state-of-art membranes (including GO membrane, metal-organic framework (MOF) membrane, polymeric membrane, silica membrane and mixed matrix membranes) in the literatures (Fig. 4c and Supplementary Table 3)[4,29–44].

The water contact angle of the TpTG$_{Cl}$@CNFs-5/PAN membrane was measured to be 7.2° and decreased to ~0° within only 0.2 s (Supplementary Fig. 25). The superhydrophilicity

might have come from the synergistic effect of both high hydrophilicity and water uptake of TpTG$_{Cl}$ and CNFs. To verify this hypothesis, the surface roughness of the membranes and the water adsorption capacity of TpTG$_{Cl}$ and CNFs were measured as shown in Supplementary Figs. 26 and 27. The TpTG$_{Cl}$/PAN membrane shows a low average surface roughness ($R_a$) of 14.8. When a droplet of water was brought in contact with the surface of the membrane, the advancing water contact angle appeared to be 29.3°. The water droplet quickly spread and permeated into the membrane within 0.2 s. The ultrafast water permeation can be explained by the capillary effect of the membrane according to the Laplace theory (Equation (1)):

$$P_c = \Delta p = -2\gamma\cos\theta_\alpha/R, \tag{1}$$

where $\Delta p$ is the liquid intrusion pressure, $\gamma$ is the surface tension of water in air, $\theta_\alpha$ is the advancing contact angle of the water on the surface, $R$ is the equivalent pore radius of the membrane. Since the $\theta_\alpha$ is <90°, the calculated $P_c$ is <0. Meanwhile, the small pore size (~1.3 nm) of TpTG$_{Cl}$/PAN membrane results in a large absolute value of $P_c$, which means that water is easy to permeate into the membrane with the capillary driving pressure of $P_c$. The above analyses can well explain the fast water spreading ability. The CNFs/PAN membrane demonstrated a hydrophilic surface with an advancing water contact angle of 31.2°, which can be attributed to the high water adsorption capacity of CNFs (675 cm$^3$ g$^{-1}$ at $P/P_0 = 0.9$) and the rough surface ($R_a$ of 66.3). However, the fast water spreading phenomenon was not observed for CNFs/PAN membrane which could be because the water permeation in CNFs was much slower than that in TpTG$_{Cl}$. The TpTG$_{Cl}$@CNFs-5/PAN membrane shows a much smaller advancing water contact angle (7.2°) than the CNFs/PAN membrane (31.2°) and TpTG$_{Cl}$/PAN membrane (29.3°). There is no significant difference in surface roughness between the CNFs/PAN membrane and TpTG$_{Cl}$@CNFs-5/PAN membrane. Therefore, the low water contact angle for the TpTG$_{Cl}$@CNFs-5/PAN membrane could be due to the synergy between TpTG$_{Cl}$ and CNFs. The TpTG$_{Cl}$@CNFs-5/PAN membrane integrates with both high water adsorption of CNFs and fast water spreading ability of TpTG$_{Cl}$, thus acquiring the smallest water contact angle. The superhydrophilic surface along with well-organized channels of the TpTG$_{Cl}$@CNFs-5/PAN membranes could remarkably lower the resistance for water permeation based on the solution-diffusion mechanism[8,45]. Furthermore, the suitable interlamellar equivalent micropore size (0.45 nm) endows the membranes with molecular sieving properties, which substantially prevents n-butanol molecules (kinetic diameter of 0.51 nm) from passing through the membrane.

A possible transport mechanism for water and n-butanol molecules across the membrane is depicted in Fig. 4d. Compared with n-butanol molecules, water molecules in the feed could be preferentially adsorbed on the superhydrophilic membrane surface. According to the water vapor adsorption-desorption isotherms in Supplementary Fig. 27, the water adsorption amount of TpTG$_{Cl}$ is higher than that of CNFs at a low relative pressure, thus the TpTG$_{Cl}$ on the surface could contribute more towards water adsorption than the CNFs. The feed would then rapidly transport through the well-organized channels of TpTG$_{Cl}$ in through-plane direction. The CNFs in the interlayer and the next TpTG$_{Cl}$ nanosheets can concentrate the water molecules to further enhance the solubility selectivity for water. Meanwhile, the interlamellar micropores with molecular sieving effects enhance the diffusion selectivity for water. The concentrated feed once again rapidly transports to the next interlayer through the nanopores in TpTG$_{Cl}$. After the repeated concentrating-sieving process through the membrane, a high permeation flux along with a high water concentration in permeate is thus

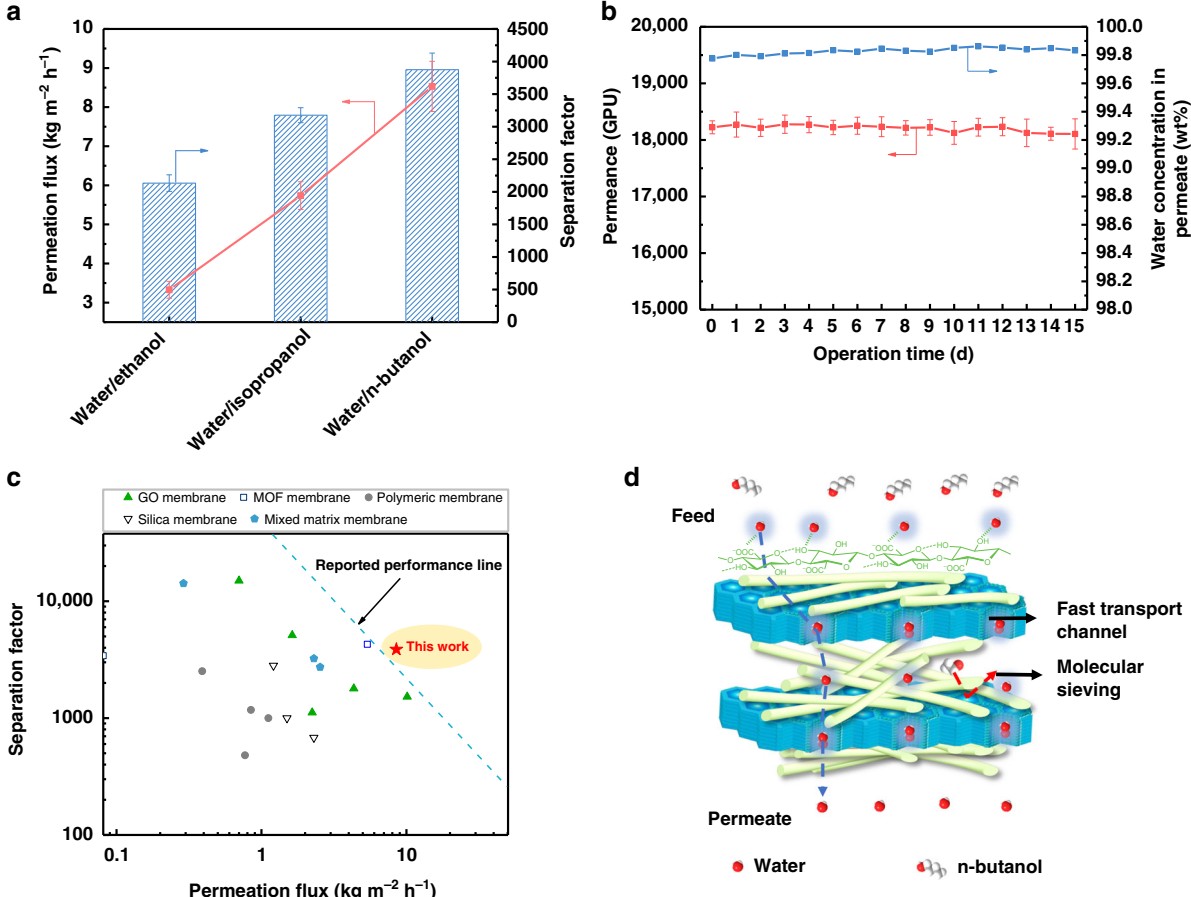

**Fig. 4** Water-selective permeation properties of the membranes. **a** Separation performance of the TpTG$_{Cl}$@CNFs-5/PAN membrane with respect to different water/alcohol systems (water concentration: 10 wt%). **b** Long-term separation performance of the membrane applied in a water/n-butanol system (water concentration: 10 wt%; operating temperature: 80 °C). **c** Comparison of the water/n-butanol separation performance of the TpTG$_{Cl}$@CNFs-5/PAN membrane and the representative membranes in the literatures. **d** A schematic diagram of the transport behavior of liquid molecules across the membrane. Error bars represent standard deviations for 3 measurements

achieved. The fast water transport of TpTG$_{Cl}$ was proved by MD simulations as shown in Supplementary Fig. 28, and a high confined water diffusion coefficient of $2.27 \times 10^{-7}$ m$^2$ s$^{-1}$ in TpTG$_{Cl}$ was calculated. To gain insight into the synergistic effect of TpTG$_{Cl}$ and CNFs, the pristine TpTG$_{Cl}$/PAN and pristine CNFs/PAN membranes were also fabricated and used for control experiments. As shown in Supplementary Fig. 29, the pristine TpTG$_{Cl}$/PAN membrane exhibits a high permeation flux but a very low separation factor due to the lack of molecular sieving. The pristine CNFs/PAN membrane exhibits both lower permeation flux and separation factor than those of the TpTG$_{Cl}$@CNFs-5/PAN membrane. Furthermore, another kind of membrane, denoted as (TpTG$_{Cl}$ + CNFs)/PAN, was prepared by directly filtering the mixtures of individual TpTG$_{Cl}$ and CNFs and used for control experiments. Its separation performance was also not desirable due to the disordered molecular transport channels (Supplementary Fig. 30). It can be deduced that the mixed-dimensional architecture not only endows the membranes with high mechanical strength, but also renders the membranes fast water-selective permeation properties.

**Dye and salt rejection performance of the membranes**. The previously reported works have indicated that the free-standing COF films show promise for nanofiltration applications[9,46]. Therefore, the application range of the TpTG$_{Cl}$@CNFs-X/PAN membranes was further evaluated by nanofiltration. The pure

water flux is nearly linear to the driving pressure, indicating the highly compact resistance of the membranes (Fig. 5a). The dye rejection performance of the membranes was evaluated by four dye solutions (congo red, alcian blue, methyl blue and orange GII) with a concentration of 100 ppm (Fig. 5b and Supplementary Figs. 31 and 32). The TpTG$_{Cl}$@CNFs-3/PAN membrane can reject congo red, alcian blue, methyl blue and orange GII with the rejection of 99.6, 98.3, 90.3, and 90.3%, respectively. The results demonstrate that the dyes with a size larger than 1 nm can be easily retained by the TpTG$_{Cl}$@CNFs-3/PAN membrane with an interlamellar equivalent micropore size of 0.82 nm, and the permeance is higher than 70 L m$^{-2}$ h$^{-1}$ bar$^{-1}$. Compared with the TpTG$_{Cl}$@CNFs-3/PAN membrane, the membranes with fewer CNFs have a lower rejection for dyes due to the larger interlamellar pore size. Although the membranes with more CNFs also have a high rejection for dyes, their permeance is much lower due to the densely packed CNF networks. Furthermore, the TpTG$_{Cl}$@CNFs-3/PAN membrane shows long-term stability up to 150 h for rejecting congo red as shown in Supplementary Fig. 33.

The salt rejection performance of the membranes was also evaluated as shown in Supplementary Fig. 34 and Fig. 5c. The TpTG$_{Cl}$@CNFs-4/PAN membrane with an interlamellar equivalent micropore size of 0.65 nm exhibits a high rejection of 96.8% as well as high permeance of 42.8 L m$^{-2}$ h$^{-1}$ bar$^{-1}$, for Na$_2$SO$_4$. The hydration radii of Cl$^-$ (0.66 nm), Na$^+$ (0.72 nm),

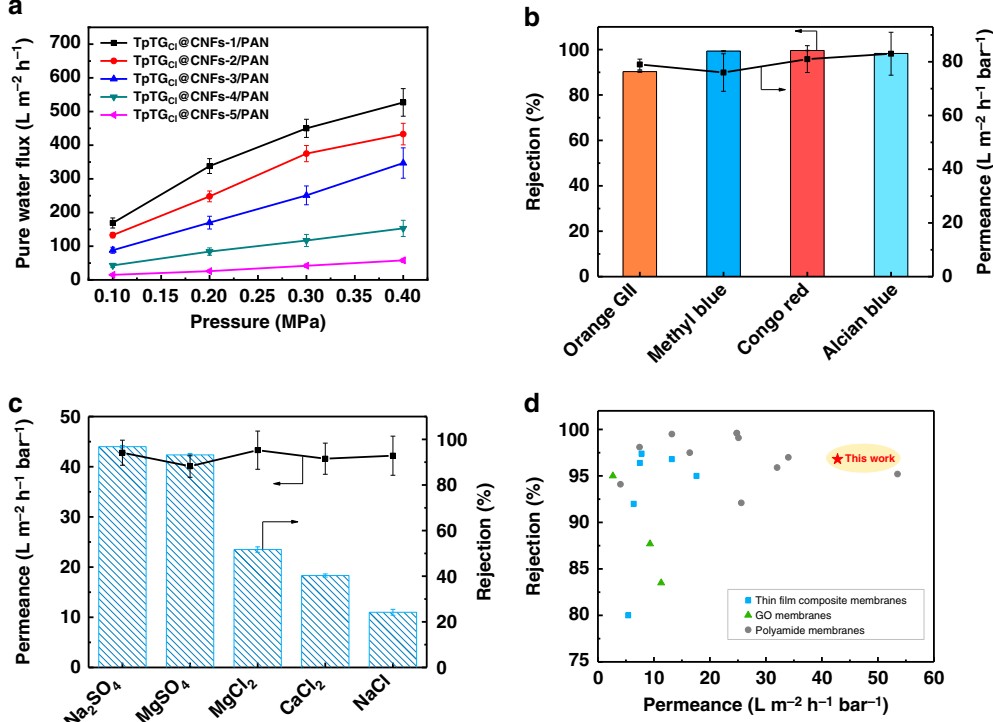

**Fig. 5** Dye and salt rejection performance of the membranes. **a** Pure water flux of the TpTG$_{Cl}$@CNFs-X/PAN membranes as a function of pressure. **b** Permeance and rejection of the TpTG$_{Cl}$@CNFs-3/PAN membrane for the nanofiltration of different dye solutions (dye concentration: 100 ppm; applied pressure: 0.2 MPa). **c** Permeance and rejection of the TpTG$_{Cl}$@CNFs-4/PAN membrane for the nanofiltration of different salt solutions (salt concentration: 1000 ppm; applied pressure: 0.4 MPa). **d** Comparison of the permeance and rejection of the TpTG$_{Cl}$@CNFs-4/PAN membrane and the representative membranes in the literatures for rejecting Na$_2$SO$_4$. Error bars represent standard deviations for 3 measurements

SO$_4^{2-}$ (0.76 nm), Ca$^{2+}$ (0.82 nm) and Mg$^{2+}$ (0.86 nm)[47] are all larger than the pore size of TpTG$_{Cl}$@CNFs-4/PAN membrane. However, due to the broad pore size distribution, the ion rejection for MgCl$_2$, CaCl$_2$ and NaCl is not high enough. The rejection mechanism could be explained by a synergistic effect of size sieving and Donnan exclusion caused by the charged groups on the membrane surface to exclude the ions with the same charge. The cations could be attracted by the negatively charged membrane surface, which would lead to a concentration difference of ions in the solution and across the membrane, and a chemical potential difference would generate[48]. The Donnan exclusion effect for sulfate salts is more notable than that for chloride salts. Furthermore, the hydration radius of Cl$^-$ (0.66 nm) is smaller than that of SO$_4^{2-}$ (0.76 nm). Driven by this Donnan exclusion combined with steric hindrance, a high rejection for sulfate salts and a relatively low rejection for chloride salts are obtained. With the increase in CNF content, the amount of negatively charged groups on membrane surface increases while the interlamellar micropore size decreases. Therefore, the rejection for salts increases while the permeance decreases with the increase in CNF content. The high permeance for the TpTG$_{Cl}$@CNFs-4/PAN membrane is mainly attributed to the abundant water transport channels of TpTG$_{Cl}$. Furthermore, the TpTG$_{Cl}$@CNFs-4/PAN membrane also shows long-term stability up to 150 h for rejecting Na$_2$SO$_4$ as shown in Supplementary Fig. 35.

As shown in Supplementary Figs. 36 and 37, the pristine TpTG$_{Cl}$/PAN membrane shows a rejection of 85.7% for congo red and a rejection of 26.2% for Na$_2$SO$_4$, which are both lower than that of the TpTG$_{Cl}$@CNFs-X/PAN membrane (the highest congo red rejection of 99.6% and the highest Na$_2$SO$_4$ rejection of 96.8%). The high permeance but low rejection of the

pristine TpTG$_{Cl}$/PAN membrane is due to the large intrinsic pore size (1.3 nm) of TpTG$_{Cl}$. The pristine CNFs/PAN membrane shows a rejection of 88.1% for congo red and a rejection of 62.4% for Na$_2$SO$_4$, which illustrates that the CNFs alone cannot form appropriate pore size for effective dye or salt rejection. Furthermore, the (TpTG$_{Cl}$ + CNFs)/PAN membrane also exhibits inferior performance than that of the TpTG$_{Cl}$@CNFs-X/PAN membrane due to the disordered molecular transport channels.

Compared with the performance of the representative nanofiltration membranes in the literatures (Fig. 5d and Supplementary Table 4)[48–66], the performance of TpTG$_{Cl}$@CNFs-4/PAN membrane is comparable to most of the representative membranes for rejecting Na$_2$SO$_4$, and it can be applied at a much lower pressure. With regard to the rejection behavior for NaCl, the as-prepared TpTG$_{Cl}$@CNFs/PAN membranes show a high ion selectivity to monovalent ions and divalent ions, demonstrating their potential application in salt rejection of solutions containing high concentration multivalent salts.

## Discussion

We described the fabrication of COF membranes through a mixed-dimensional assembly of 2D COF nanosheets and 1D CNFs. This unique design resulted in the fast and highly selective water transport as well as the high stability of the membranes. This assembly strategy can also be applied to other 2D COFs to fabricate robust COF-based membranes. To further validate this idea, another four kinds of Schiff-base-type COFs i.e. TpPa-1, TpBD, TpHZ and TpBD(OH)$_2$[67–69] were synthesized by Tp and p-phenylenediamine (Pa-1), benzidine (BD), hydrazine hydrate (HZ), or 3,3'-dihydroxybenzidine (BD(OH)$_2$). The molecular structures and morphology of these COFs are shown in

Supplementary Fig. 38. The structures of these COFs were confirmed by PXRD, [13]C CP-MAS solid-state NMR and FTIR as shown in Supplementary Figs. 39 and 40. These COF nanosheets were also assembled with CNFs to form stable COFs@CNFs nanocomposites by using the same method. Additional four kinds of membranes including TpPa-1@CNFs/PAN, TpBD@CNFs/PAN, TpHZ@CNFs/PAN and TpBD(OH)$_2$@CNFs/PAN were fabricated and these membranes showed dense and defect-free structures (Supplementary Fig. 41). These membranes also display significantly improved mechanical strength compared with the corresponding pristine COF membrane as shown in Supplementary Table 5. Furthermore, these membranes could also exhibit high permeance as well as high rejection for dyes and salts, as shown in Supplementary Table 6.

In summary, we have explored a mixed-dimensional assembly strategy to design advanced COF membranes. The strategy of combining nanoporous 2D COFs with flexible and ultrastrong 1D CNFs solves the current separation performance-mechanical strength dilemma in COF membrane fabrication and its effective application for molecular separations. High separation performance of the membranes has been achieved in alcohol dehydration, dye rejection and salt rejection. The mechanical strength of the membranes has also been improved owing to the multiple interactions. The designable structures and extremely high stability of COFs offer an intriguing opportunity to develop advanced synthetic membranes. The mixed-dimensional design in our study can apply to fabricate a series of various COF membranes. These COF membranes have demonstrated high potentials in energy and environment-relevant separation processes such as solvent dehydration and water reuse.

## Methods

**General**. The experimental materials and detailed synthesis procedure of COFs (TpTG$_{Cl}$, TpPa-1, TpBD, TpHZ, and TpBD(OH)$_2$) are given in the Supplementary methods.

**Preparation of the TpTG$_{Cl}$@CNFs-X nanocomposites**. The TpTG$_{Cl}$ dispersion and CNF dispersion were prepared by using deionized water (pH = 6.0). To fabricate the TpTG$_{Cl}$@CNFs-X nanocomposites, a certain volume (1, 2, 3, 4, 5, 6 or 7 mL) of CNF dispersion (0.03 wt%) was added into 25 mL of TpTG$_{Cl}$ dispersion (0.5 mg mL$^{-1}$) with vigorous stirring for 12 h at 60 °C. In this procedure, stable mixed-dimensional TpTG$_{Cl}$@CNFs-X nanocomposites with a planar TpTG$_{Cl}$ nanosheet uniformly encapsulated into a dense network of 1D nanofibers were formed through a self-assembly. The products were centrifuged at 10,000 rpm for 15 min to remove the extra CNFs that could not bond with TpTG$_{Cl}$ surface. The resulted TpTG$_{Cl}$@CNFs-X nanocomposites were dialyzed against deionized water for 5 d by a dialysis membrane tube with a molecular weight cut-off of 12,000, resulting in the stable dispersion of TpTG$_{Cl}$@CNFs-X nanocomposites.

**Preparation of additional COFs@CNFs-X nanocomposites**. Additional four kinds of mixed-dimensional nanocomposites including TpPa-1@CNFs-X, TpBD@CNFs-X, TpHZ@CNFs-X and TpBD(OH)$_2$@CNFs-X were also prepared by the same procedure as described above.

**Preparation of the membranes**. The PAN substrates were immersed in water for 2 d to remove any residues in the pores prior to use. The TpTG$_{Cl}$@CNFs-X/PAN membranes were fabricated by filtering a certain volume (5, 10, 30, 100, 250, or 500 mL) of the TpTG$_{Cl}$@CNFs-X solution (0.005 mg mL$^{-1}$) on PAN substrates through a vacuum-assisted self-assembly method. The COFs@CNFs nanocomposites were stacked into membranes in one-step driven by the pressure and then the membranes were dried at 40 °C for 24 h. The thickness of TpTG$_{Cl}$@CNFs-X membrane can be tuned by varying the volume of filtrate. The pristine TpTG$_{Cl}$/PAN and pristine CNFs/PAN membranes were fabricated for comparison by filtering the alone TpTG$_{Cl}$ solution or alone CNF solution on PAN substrates, in which the contents of TpTG$_{Cl}$ and CNFs were equal to the corresponding TpTG$_{Cl}$@CNFs-X/PAN membranes. The (TpTG$_{Cl}$ + CNFs)/PAN membranes were also fabricated for comparison by directly filtering the mixtures of TpTG$_{Cl}$ and CNFs on PAN substrates without a pre-assembly, in which the contents of TpTG$_{Cl}$ and CNFs were also equal to the corresponding TpTG$_{Cl}$@CNFs-X/PAN membranes. The TpPa-1@CNFs-X/PAN, TpBD@CNFs-X/PAN, TpHZ@CNFs-X/PAN and TpBD(OH)$_2$@CNFs-X/PAN membranes were also prepared by the same

procedure. For each type of membrane, at least three samples were prepared and tested to ensure the experimental reproducibility.

**Characterization**. TEM images and element distribution mappings were taken by a JEM-2100F device, the FFT and IFFT patterns were obtained from the Gatan DigitalMicrograph Software. XRD analysis was conducted by a D/MAX-2500 instrument with Cu-Kα radiation. AFM images were observed by a multifunctional scanning probe microscope (NTEGRA Spectra). Zeta potentials were measured by a Nano ZS instrument with a 4 mW He-Ne laser. N$_2$ adsorption-desorption isotherms were collected by a Quantachrome Autosorbe-1 analyzer at 77 K, the surface area and pore size distribution were calculated by BET and NL-DFT methods. SEM images were observed by a Nanosem 430 device. FTIR was carried out by a BRUKER Vertex 70 equipment. XPS was performed on an ESCALAB 250Xi instrument by using an Al-Kα radiation source. Solid-state CP/MAS NMR spectrum was collected by a 300 MHz Varian InfinityPlus spectrometer. Water vapor adsorption-desorption isotherms were collected by a VTI-SA+ steam adsorption instrument at 298 K. Water contact angles were tested by a JC2000D2M static contact angle goniometer (POWEREACH®). TGA curves were collected by the NETZSCH 209 F3 equipment under a N$_2$ atmosphere. The mechanical properties of the membranes were measured by an electronic tensile machine (WDW-2) at a stretching rate of 1 mm min$^{-1}$.

**Water-selective permeation measurements**. The water-selective permeation properties of the membranes were performed by a homemade pervaporation membrane module. The effective membrane area was 8.5×10$^{-4}$ m$^2$. At the upstream side of the membrane module, the feed mixture was circulated at a rate of 60 L h$^{-1}$ by a gear pump, and the pressure at the downstream side was kept below 0.3 kPa by using a vacuum pump. The vapor permeate was collected by a cold trap inserted in liquid N$_2$ and weighed after a certain period of time. The products were analyzed by a gas chromatography (Agilent 7820A). The permeation flux (J, kg m$^{-2}$ h$^{-1}$) and separation factor (α) were calculated by Eqs. (2) and (3):

$$J = \frac{M}{S \times t}, \tag{2}$$

$$\alpha = \frac{P_W/(1 - P_W)}{F_W/(1 - F_W)}, \tag{3}$$

where M represents the mass of permeate (kg), S is the membrane area (m$^2$), and t refers to the time interval (h). $F_W$ and $P_W$ are the mass fraction of water in feed and permeate, respectively.

By eliminating the influence of partial pressure driving force, the permeance ((P/l), GPU) of the individual component was calculated by Equation (4) and the selectivity (β) was calculated by Equation (5).

$$(P/l)_i = \frac{J_i}{p_{i0} - p_{il}} = \frac{J_i}{\gamma_{i0} x_{i0} p_{i0}^{sat} - p_{il}}, \tag{4}$$

$$\beta = \frac{(P/l)_W}{(P/l)_A}, \tag{5}$$

where l refers to the membrane thickness (m). For individual component i, $J_i$ is the permeation flux (kg m$^{-2}$ h$^{-1}$), $p_{i0}$ and $p_{il}$ are the partial pressure (Pa) in the upstream side and the downstream side, respectively. $\gamma_{i0}$ refers to the activity coefficient which is calculated by the Wilson equation. $x_{i0}$ is the mole fraction in the feed and $p^{sat}_{i0}$ is the saturated vapor pressure obtained through the Antoine equation. Subscript W and A stand for water and alcohol, respectively.

**Nanofiltration performance test**. The separation test was carried out on a dead-end vacuum filtration device with an effective membrane area of 4.9×10$^{-4}$ m$^2$. Before testing, the membrane was compacted for more than 30 min at 0.2 MPa to reach a steady state. The flux (J, L m$^{-2}$ h$^{-1}$) was calculated by Eq. (6), where the density (ρ) of permeate is considered to be 1 g cm$^{-3}$ due to the low dye/salt concentration. Permeance (P, L m$^{-2}$ h$^{-1}$ bar$^{-1}$) and rejection (R, %) of dyes or salts were calculated by Equation (7) and (8), where Δp is the applied pressure, and $C_P$ and $C_f$ are the solute concentration in permeate and feed, respectively. UV-vis spectrophotometer (Hitachi UV-2800) and electrical conductivity (Leichi DDS-11A) were used to determine the dye concentration and salt concentration, respectively.

$$J = \frac{M}{\rho \times S \times t}, \tag{6}$$

$$P = \frac{J}{\Delta p}, \tag{7}$$

$$R = \left(1 - \frac{C_P}{C_f}\right) \times 100\%. \tag{8}$$

**MD simulations**. The MD simulations were performed by the Material Studio software. The models of TpTG$_{Cl}$ and CNFs were established based on the previous

studies[34,70]. The structures of TpTG$_{Cl}$ and CNFs were optimized using the COMPASS force field and the non-bonded interaction energies were calculated based on a Dreiding force field. The non-bonded interactions include van der Waals forces, hydrogen bonds and electrostatic interactions. The van der Waals energy $E_{vdW}$ and the hydrogen bond energy $E_{H-bond}$ are described by the Lennard–Jones (12–6) and Lennard–Jones (12–10) potential, respectively. The electrostatic energy $E_{Electrostatic}$ is described by a screened (distance-dependent) Coulombic term and atomic monopoles. The non-bonded interaction energy $\Delta E$ between TpTG$_{Cl}$ and CNFs was calculated by Equation (9):

$$\Delta E = E_{TpTG_{Cl}@CNFs} - (E_{TpTG_{Cl}} + E_{CNFs}), \qquad (9)$$

where $E_{TpTG_{Cl}@CNFs}$ is the non-bonded energy of TpTG$_{Cl}$@CNFs system, $E_{TpTG_{Cl}}$ and $E_{CNFs}$ is the non-bonded energy of TpTG$_{Cl}$ and CNFs, respectively.

For the simulation of the water permeation through the TpTG$_{Cl}$ nanosheets, a thousand water molecules were placed in the chamber on one side (along the z direction) of the TpTG$_{Cl}$ as the feed chamber. The vacuum chamber was placed on the other side of the TpTG$_{Cl}$ as the permeate chamber. The TpTG$_{Cl}$ nanosheet of 3.15 nm × 3.15 nm was placed in the simulated box, and a periodic boundary condition was applied to the x-y direction. The z length of the simulation box was 11.0 nm. The TpTG$_{Cl}$ nanosheets were modeled by the UFF force field with QEq charge[71]. The SPC/E model was used to describe water molecules[72]. In the diffusion simulation, the system was subjected to energy minimization using the smart minimizer method which switched from steepest-descent to conjugated gradient and then to the Newton method. Then, a 1 ns NVT (constant particle number, volume and temperature) simulation was performed with a time step of 1 fs. The Berendsen thermostat was employed to maintain the temperature of 353 K. The atoms of the TpTG$_{Cl}$ were frozen in the simulation. Driven by the concentration gradient, water molecules in the feed chamber would pass through the pores of the TpTG$_{Cl}$ to the permeate chamber. The diffusion coefficients were calculated using the Einstein relation (Equation (10)):

$$D = \lim_{t \to \infty} \frac{1}{6t} \left( \frac{1}{N} \sum_{N}^{k=1} |r_k(t) - r_k(0)|^2 \right), \qquad (10)$$

where $N$ is the number of molecules and $r_k(t)$ is the position of the $k$th molecule at time $t$.

## Data Availability
The source data underlying Figs. 1d, e, h, 3b, c, 4a, b and 5a–c are provided as a Source Data file. The data that support the findings of this study are available from the corresponding author upon reasonable request.

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

## Acknowledgements

This research was supported by the National Natural Science Foundation of China (No. 21490583, 21621004, 21878215, 21878216, 21576189), the State Key Laboratory of Organic-Inorganic Composites (No. oic-201801003).

## Author contributions

H.Y., F.P., and Z.J. devised the idea. Z.X., Y.Z and Y.L. performed the experiments. H.W., L.Y., H.J.W., N.N. and Y.S. analyzed the data. H.Y., H.W., F.P., and Z.J. co-wrote the manuscript. All authors participated in the discussion and commented on the results.

## Additional information

**Competing interests:** The authors declare no competing interests.

