## [Peer Review File · Nature Communications]

Reviewers' comments:

Reviewer #1 (Remarks to the Author):

In this manuscript, Jiang and coworkers describe their work on preparing composite membranes containing alternating layers of β -ketoenamine linked covalent organic frameworks (COFs) and cellulose nanofibers (CNFs) on top of a polyacrylonitrile (PAN) support for use in chemical separations, including removal of alcohols, organic dyes, and sodium sulfate from water. This work is unique in that the co-assembly of COF nanosheets and CNFs is explored for the first time. Therefore, this work will be of an important contribution to the growing body of COF literature. However, in its current form, the manuscript suffers from an overall lack of clarity, and several important control experiments are missing. Furthermore, the authors indicate that they prepared membranes from many different COFs in their methods section, but these materials are never mentioned in the body of the manuscript, nor is any characterization data included for them outside of the supporting information. Additionally, the authors repeatedly state that their materials exhibit improved separation performance compared to COFs alone, but they do not have any data (references) or control experiments to support these assertions. Therefore, this manuscript should not be published in Nature Communications.

Major concerns:

1. Proof read. There are several grammatical errors that reduce the clarity of the manuscript.
2. Reproducibility. The authors include some vague statements about how these materials are synthesized, but there are no specific instructions included that another scientist may follow to reproduce these materials. These must be included.
3. The authors do not clarify how they achieve multiple layers of COF interspersed with CNFs. Is the synthesis conducted repetitively to achieve an optimal number of layers? If multiple layers are made in one batch, how is this controlled? The authors state they can tune the thickness of the layer by varying the volume of filtrate, but does this alter the thickness of one layer? This point goes along with the reproducibility needed in point 2.
4. The authors make several large claims, like that their materials exhibit improved mechanical properties and improved separation performance compared to freestanding COF films. However, the authors include no direct comparison or reference to the characterization and performance of freestanding COF films. Control experiments should be included that compare the dye and salt rejection properties of the TpTGCI@CNFs-X membranes to the COF and CNFs alone.
5. The authors have not included critical references for this manuscript, and these references indicate that free standing COF films do show an impressive ability for separations chemistry. Among the many references from the last 12 months, the authors should include Banerjee's 2017 manuscript from J. Am. Chem. Soc. and Dichtel's 2018 manuscript from Chem. The data points for thin film composite membranes shown in Figure 5d also lack references in the figure caption and the manuscript text.
6. The figures should be redone. Many of them are blurry or otherwise illegible and lack appropriate labels such that they are difficult to understand. For example, Fig. 2a contains several unlabeled structures that make this depiction of the vacuum-assisted self-assembly process difficult to interpret. Furthermore, the SEM images contain scale bars with no labels, which should be corrected.
7. The authors should justify why they chose to use the TpTGCI COF for their experiments as opposed to any other COF.
8. The authors should clarify what COF morphology is obtained from the TpTGCI synthesis. Is this material isolated as a microcrystalline powder, a suspension of nanoplatelets, or a freestanding film?
9. The authors should clarify which characterization data pertains to the TpTGCI COF alone, the CNFs alone, and the TpTGCI@CNFs-X membranes. At times, the authors appear to switch from discussing one to another without indicating that they are doing so.
10. The authors indicate that the large (001) peak in the PXRD spectrum of their TpTGCI indicates that the COF is horizontally oriented. However, these data alone do not provide enough evidence to conclude that the film is oriented, just that there is stacking of the COF pores, but not in which direction this stacking occurs. Additional grazing-incidence wide-angle X-ray scattering (GIWAXS) data is required.
11. The authors should explain the term "Donnan exclusion".

12. The authors mention in the "Methods" section that several other COF-CNF composite membranes were prepared from a variety of COFs, but the acronyms used to describe these materials are never defined and no characterization data is included for these alternative structures outside of the supporting information. These should be discussed in the manuscript.

13. The authors should rationalize why some TpTGCl@CNFs-X membranes perform better in certain applications (solvent dehydration, dye rejection, and desalination) than others with more or less CNFs.

14. The authors should more thoroughly explain why Na₂SO₄ is rejected by the TpTGCl@CNFs-4 membrane but the chloride salts are not.

Reviewer #2 (Remarks to the Author):

The development of covalent organic framework (COF) membranes for molecular separation has become a hot topic in membrane science and technology in recent years. This paper reports COFs@CNFs composites as a membrane material for possible applications in butanol pervaporation dehydration as well as dye separation and desalination. The characterization of the COFs@CNFs composite material is excellent, and the composite membranes indeed show improved performance in comparison with COF membranes. However, the concept for preparation of the composite membranes with addition of CNFs is quite common, which has been widely reported in the literatures for various membrane materials. Therefore, taking this point into consideration, the novelty of this paper remains an important concern for publication in this journal.

1. Compared with the water contact angles of PAN, TpTGCl/PAN and CNFs/PAN, why the water contact angle of TpTGCl@CNFs-5/PAN is much smaller? The authors should clearly explain the relationship between these properties and the material structures, which would be helpful for understanding water transport mechanism of the composite membrane.

2. In addition to pore size distribution, what is average pore size of the membranes? Which is very important to understand the transport behaviors of the membranes.

3. The long-term stability of TpTGCl@CNFs-5/PAN membrane seems a big concern, why the water permeation flux continuously decreases with operating time for butanol dehydration?

4. There is no information about long-term stability of dye separation and desalination performance. It is better to add additional experiments to confirm the stability in above systems, since the membrane is not so stable for butanol dehydration.

5. Regarding the membrane desalination performance, although the COFs@CNFs composite membranes show a high permeation flux and acceptable Na₂SO₄ rejection, but the rejection for CaCl₂ and NaCl is quite low. As a matter of fact, so many membranes have been reported for desalination with both high CaCl₂ and NaCl rejection. The comparison of desalination performance should involve the CaCl₂ and NaCl systems in order to objectively judge the COFs@CNFs composite membrane performance for desalination application.

Reviewer #3 (Remarks to the Author):

Overall, this paper is in its current state quite confusing. Many (crucial) aspects are unclear. Performance nor preparation method seem at this moment good enough to me to make serious impact.

I find the link with 'bio-inspired' a bit far-fetched. I understand it is a popular selling point, but here I find it a bit exaggerated for use. The honeycomb is just a direct consequence of the chosen building blocks, no?

Is the Tyndall effect relevant enough to mention here so explicitly?

Is this really self-assembly? It is only the COF that self-assembles, right? Not the full membrane? Better schemes are required to explain the transport. Is it only the interlamellar micropores that are relevant for the transport? Not the ring in the COF?

SuppTable 4: Quite sure better PA-membranes have been reported in terms of selectivity for sure.

Pressures are all so low, far below real desalination!!

I could not find how the selectivity was calculated in PV.

The role of Donnan exclusion: it is explained like ions are attracted to the membrane. This is undesired, no?

In addition, language is often not 100% correct or clear.

E.g. First line second paragraph in abstract (I guess it is referring to porous materials only?)

Line 43: 'greater than': shouldn't this be the case for permeance?

64: what is meant with a 'sheltering effect'?

Minor:

Refs sometimes contain 'et al.'

Many figures are not clear, especially when there is text in them, which can not be well read.

Response to the reviewers' comments

Reviewer #1

In this manuscript, Jiang and coworkers describe their work on preparing composite membranes containing alternating layers of β -ketoenamine linked covalent organic frameworks (COFs) and cellulose nanofibers (CNFs) on top of a polyacrylonitrile (PAN) support for use in chemical separations, including removal of alcohols, organic dyes, and sodium sulfate from water. This work is unique in that the co-assembly of COF nanosheets and CNFs is explored for the first time. Therefore, this work will be of an important contribution to the growing body of COF literature. However, in its current form, the manuscript suffers from an overall lack of clarity, and several important control experiments are missing. Furthermore, the authors indicate that they prepared membranes from many different COFs in their methods section, but these materials are never mentioned in the body of the manuscript, nor is any characterization data included for them outside of the supporting information.

Additionally, the authors repeatedly state that their materials exhibit improved separation performance compared to COFs alone, but they do not have any data (references) or control experiments to support these assertions. Therefore, this manuscript should not be published in Nature Communications.

Response: Thanks this reviewer for the valuable guidance on our manuscript. Sorry for the lack of clarity and missing of some important control experiments. We have gone through and revised the manuscript carefully to make it much clearer. We have taken the effort to improve the clarity and have also supplemented important control experiments and characterization data in the revised manuscript. The separation performance and mechanical strength of pristine COF membrane and pristine CNF membrane have been added to support the assertions that the COFs@CNFs membranes exhibit remarkably improved performance. The characterization data of other four kinds of COFs@CNFs membranes have been added and discussed in the revised manuscript.

Major concerns:

1. Proof read. There are several grammatical errors that reduce the clarity of the manuscript.

Response: We have carefully checked the grammatical errors and tried our best to improve the language. We hope the language now can better meet the high standards for publication.

For example:

Full text: “COFs membranes” has been revised to “**COF membrane**”.

Page 3, line 8: “greater than” has been revised to “**larger than**”.

Page 4, line 11: “entrance pore size” has been revised to “**size of pore entrance**”.

Page 6, line 6: “considered to integrate with” has been revised to “**considered to have been integrated with**”.

Page 6, line 8: “at different pH” has been revised to “**at different pH values**”.

Page 6, line 8: “forms” has been revised to “**are formed**”.

Page 10, line 21: “With further loading” has been revised to “**On further stretching**”.

Page 20, line 2: “applicable” has been revised to “**applied**”.

Page 20, line 2-3: “To demonstrate” has been revised to “**To further validate this novel idea**”.

Page 20, line 17: “The strategy that” has been revised to “**The strategy of**”.

Page 20, line 21: “also improves” has been revised to “**has also been improved**”.

2. Reproducibility. The authors include some vague statements about how these materials are synthesized, but there are no specific instructions included that another scientist may follow to reproduce these materials. These must be included.

Response: According to the reviewer’s comment, more detailed information has been added in the experimental section as listed below.

General. The experimental materials and detailed synthesis procedure of COFs (TpTG_{Cl}, TpPa-1, TpBD, TpHZ, TpBD(OH)₂) are given in the Supplementary methods.

Preparation of the TpTG_{Cl}@CNFs-X nanocomposites. The TpTG_{Cl} dispersion and CNF dispersion were prepared by using deionized water (pH=6.0). To fabricate the TpTG_{Cl}@CNFs-X nanocomposites, a certain volume (X=1, 2, 3, 4, 5, 6, 7 mL) of CNF dispersion (0.03 wt%) was added into 25 mL of TpTG_{Cl} dispersion (0.5 mg mL⁻¹) with vigorous stirring for 12 h at 60 °C. In this procedure, stable mixed-dimensional TpTG_{Cl}@CNFs-X nanocomposites with a planar TpTG_{Cl} nanosheet uniformly encapsulated into a dense network of 1D nanofibers were formed through a self-assembly. The products were centrifuged at 10000 rpm for 15 min to remove the extra CNFs that could not bond with TpTG_{Cl} surface. The resulted TpTG_{Cl}@CNFs-X nanocomposites were dialyzed against deionized water for 5 d by a dialysis membrane tube with a molecular weight cut-off of 12000, resulting in the stable dispersion of TpTG_{Cl}@CNFs-X nanocomposites.

Preparation of the TpPa-1@CNFs-X, TpBD@CNFs-X, TpHZ@CNFs-X and TpBD(OH)₂@CNFs-X nanocomposites. Additional four kinds of mixed-dimensional nanocomposites including TpPa-1@CNFs-X, TpBD@CNFs-X, TpHZ@CNFs-X and TpBD(OH)₂@CNFs-X were also prepared by the same procedure as described above.

Preparation of the membranes. The PAN substrates were immersed in water for 2 d to remove any residues in the pores prior to use. The TpTG_{Cl}@CNFs-X/PAN membranes were fabricated by filtering a certain volume (5, 10, 30, 100, 250, or 500 mL) of the TpTG_{Cl}@CNFs-X solution (0.005 mg mL⁻¹) on PAN substrates through a vacuum-assisted self-assembly method. The

COFs@CNFs nanocomposites were stacked into membranes in one-step driven by the pressure and then the membranes were dried at 40 °C for 24 h. The thickness of TpTG_{Cl}@CNFs-X membrane can be tuned by varying the volume of filtrate. The pristine TpTG_{Cl}/PAN and pristine CNFs/PAN membranes were fabricated for comparison by filtering the alone TpTG_{Cl} solution or alone CNF solution on PAN substrates, in which the contents of TpTG_{Cl} and CNFs were equal to the corresponding TpTG_{Cl}@CNFs-X/PAN membranes. The (TpTG_{Cl} + CNFs)/PAN membranes were also fabricated for comparison by directly filtering the mixtures of TpTG_{Cl} and CNFs on PAN substrates without a pre-assembly, in which the contents of TpTG_{Cl} and CNFs were also equal to the corresponding TpTG_{Cl}@CNFs-X/PAN membranes. The TpPa-1@CNFs-X/PAN, TpBD@CNFs-X/PAN, TpHZ@CNFs-X/PAN and TpBD(OH)₂@CNFs-X/PAN membranes were also prepared by the same procedure. For each type of membrane, at least three samples were prepared and tested to ensure the experimental reproducibility.

3. The authors do not clarify how they achieve multiple layers of COF interspersed with CNFs. Is the synthesis conducted repetitively to achieve an optimal number of layers? If multiple layers are made in one batch, how is this controlled? The authors state they can tune the thickness of the layer by varying the volume of filtrate, but does this alter the thickness of one layer? This point goes along with the reproducibility needed in point 2.

Response: Sorry for the unclear statement. Actually, the COF nanosheets were initially pre-assembled with CNFs to obtain stable mixed-dimensional nanocomposites (COFs@CNFs), which consisted of a COF nanosheet uniformly encapsulated into a dense network of 1D nanofibers. Then the COFs@CNFs nanocomposites were used for fabricating membranes by a vacuum-assisted self-assembly method (Fig. 2a). In this procedure, the COFs@CNFs nanocomposites were stacked into the membrane in one-step. The concentration of the dispersion of COFs@CNFs nanocomposites was constant, thus the thickness of the membrane can be tuned by varying the volume of the filtrate. To clarify the synthesis procedure and the structures of these materials, we have revised the schematic illustration (Fig. 2a) and the description about the synthesis and structures of the membranes (**Page 21-22**).

Fig. 2a A schematic illustration showing the vacuum-assisted self-assembly and the mixed-dimensional nanostructure.

4. The authors make several large claims, like that their materials exhibit improved mechanical properties and improved separation performance compared to freestanding COF films. However, the authors include no direct comparison or reference to the characterization and performance of freestanding COF films. Control experiments should be included that compare the dye and salt rejection properties of the TpTG_{Cl}@CNFs-X membranes to the COF and CNFs alone.

Response: According to the reviewer’s valuable guidance, the control experiments including the mechanical properties of the freestanding pristine COF membrane and pristine CNF membrane, as well as the separation performance of the pristine COF and pristine CNF membranes were performed.

The mechanical properties of the pristine TpTG_{Cl} membrane, pristine CNF membrane and the TpTG_{Cl}@CNFs-X membranes are shown in Figure R1. The tensile strength and Young’s modulus of the pristine TpTG_{Cl} membrane are 15.2 MPa and 1.98 GPa, respectively. The freestanding TpTG_{Cl}@CNFs-X membranes reveal much higher tensile strength (up to 108.3 MPa) and elastic modulus (up to 3.62 GPa) compared with the freestanding pristine TpTG_{Cl} membrane and CNF membrane. These results have been added into the revised manuscript as **Fig. 3c**. The description “**The mechanical properties of the freestanding membranes are shown in Fig. 3c. The TpTG_{Cl}@CNFs-X membranes show much higher tensile strength (up to 108.3 MPa) and elastic modulus (up to 3.62 GPa) compared with the pristine TpTG_{Cl} membranes and the pristine CNF membrane.**” has been added in the revised manuscript (**Page 10, line 9-12**).

Figure R1. The mechanical properties of the pristine TpTG_{Cl} membrane, pristine CNF membrane and the TpTG_{Cl}@CNFs-X membranes.

The pristine TpTG_{Cl}/PAN, pristine CNFs/PAN and (TpTG_{Cl} + CNFs)/PAN membrane (fabricated by directly filtering the mixtures of TpTG_{Cl} and CNFs on PAN substrates without a pre-assembly) were prepared. The separation performance of these membranes for dye and salt rejection is shown in Figure R2 and R3. The pristine TpTG_{Cl}/PAN membrane shows a rejection of 85.7% for congo red and a rejection of 26.2% for Na₂SO₄, which are both lower than that of the TpTG_{Cl}@CNFs-X/PAN membrane (the highest congo red rejection of 99.6% and the highest Na₂SO₄ rejection of 96.8%). The high permeance but low rejection of the pristine TpTG_{Cl}/PAN membrane is due to the large intrinsic pore size (1.3 nm) of TpTG_{Cl}. The pristine CNFs/PAN membrane shows a rejection of 88.1% for congo red and a rejection of 62.4% for Na₂SO₄, which illustrates that the CNFs alone cannot form appropriate pore size for effective dye and salt rejection. Furthermore, the (TpTG_{Cl} + CNFs)/PAN membrane fabricated without pre-assembly also exhibits inferior performance than that of the TpTG_{Cl}@CNFs-X/PAN membrane due to the disordered transport channels. These results and discussion have been added into the revised supplementary information (**Supplementary Figure 36 and 37**) and revised manuscript (**Page 19, line 5-13**).

Figure R2. Nanofiltration performance comparison of the pristine TpTG_{Cl}/PAN membrane, pristine CNFs/PAN membrane, (TpTG_{Cl} + CNFs)/PAN membrane and TpTG_{Cl}@CNFs-3/PAN membrane for rejecting congo red.

Figure R3. Nanofiltration performance comparison of the pristine TpTG_{Cl}/PAN membrane, pristine CNFs/PAN membrane, (TpTG_{Cl} + CNFs)/PAN membrane and TpTG_{Cl}@CNFs-4/PAN membrane for rejecting Na₂SO₄.

5. The authors have not included critical references for this manuscript, and these references indicate that free standing COF films do show an impressive ability for separations chemistry.

Among the many references from the last 12 months, the authors should include Banerjee's 2017 manuscript from *J. Am. Chem. Soc.* and Dichtel's 2018 manuscript from *Chem.* The data points for thin film composite membranes shown in Figure 5d also lack references in the figure caption and the manuscript text.

Response: The following two references from Banerjee and Dichtel research groups have been added in the revised manuscript as Refs 12 and 58:

12. Dey, K. et al. Selective molecular separation by interfacially crystallized covalent organic framework thin films. *J. Am. Chem. Soc.* 139, 13083-13091 (2017).

58. Matsumoto, M. et al. Lewis-acid-catalyzed interfacial polymerization of covalent organic framework films. *Chem* 4, 308-317 (2018).

Another 21 references for the data points in Figure 5d have also been added in the revised manuscript as Refs 60-80.

6. The figures should be redone. Many of them are blurry or otherwise illegible and lack appropriate labels such that they are difficult to understand. For example, Fig. 2a contains several unlabeled structures that make this depiction of the vacuum-assisted self-assembly process difficult to interpret. Furthermore, the SEM images contain scale bars with no labels, which should be corrected.

Response: According to the reviewer's valuable suggestion, we have carefully checked all of the figures. The major changes in the figures are shown below.

The Fig. 2a has been revised:

Original Fig. 2a

Revised Fig. 2a

Other figures such as Fig. 1d-h, Fig. 4d, Supplementary Fig. 5, 8f, 11, 20, 23, and 24 have also been revised.

The scale bar labels in SEM images have not been revised in order to meet the requirement of the format of this journal: **the scale bar labels should be given in the figure caption**. The editor also advised us not to change the scale bar labels for the moment.

7. The authors should justify why they chose to use the TpTGCl COF for their experiments as opposed to any other COF.

Response: Actually, many kinds of amine-linked or azine-linked COFs can be chosen to fabricate the membranes through the mixed-dimensional assembly as shown in Supplementary Figure 41. The COF TpTGCl possesses a relatively small pore size and high stability owing to the irreversible enol-keto tautomerism. The self-exfoliated feature of TpTGCl arising from the presence of loosely bound chloride ions in the interlayer makes it easy to obtain COF nanosheets. Most importantly, the intrinsic positive charge of guanidinium units can readily interact with CNFs by the hydrogen bond and electrostatic interaction, resulting in the higher stability of both the nanocomposites and membranes than those fabricated by other COFs. According to the reviewer's suggestion, a brief explanation on why COF TpTGCl was chosen has been added in the revised manuscript: **"The COF TpTGCl can be easily exfoliated to obtain chemically stable nanosheets. Meanwhile, the intrinsic positive charge of guanidinium units on TpTGCl framework makes it a good candidate to fabricate stable nanocomposites by assembling with negatively charged components"** (Page 5, line 1-4).

8. The authors should clarify what COF morphology is obtained from the TpTGCl synthesis. Is this material isolated as a microcrystalline powder, a suspension of nanoplatelets, or a freestanding film?

Response: The synthesized COF TpTGCl is isolated as a suspension of nanosheets. The description **"The products were then dispersed into deionized water to obtain a suspension**

of TpTG_{Cl} nanosheets” has been added in the revised **supplementary information (Page 3, line 11-12)**.

9. The authors should clarify which characterization data pertains to the TpTG_{Cl} COF alone, the CNFs alone, and the TpTG_{Cl}@CNFs-X membranes. At times, the authors appear to switch from discussing one to another without indicating that they are doing so.

Response: We apologize for the unclear description of the characterization data. According to the reviewer’s valuable comment, the unclear statements in the manuscript have been revised.

For example:

Page 6, line 3-4: The original description “PXRD patterns of the as-prepared TpTG_{Cl}@CNFs-X” has been revised to “**PXRD patterns of the as-prepared TpTG_{Cl}@CNFs-X nanocomposites**”.

Page 8, line 12-14: The original description “(001) peak with high intensity in the PXRD results further...” has been revised to “**The PXRD pattern of the TpTG_{Cl}@CNFs-X membrane is shown in Supplementary Fig. 17, which also displays...**”.

10. The authors indicate that the large (001) peak in the PXRD spectrum of their TpTG_{Cl} indicates that the COF is horizontally oriented. However, these data alone do not provide enough evidence to conclude that the film is oriented, just that there is stacking of the COF pores, but not in which direction this stacking occurs. Additional grazing-incidence wide-angle X-ray scattering (GIWAXS) data is required.

Response: The GIWAXS pattern of the membrane was measured on a XEUSS SAXS/WAXS system at an incident angle of 0.2°. The GIWAXS pattern of the TpTG_{Cl}@CNFs-X membrane is shown in Figure R4, which exhibits two peaks. The peak with a q_z of 17.8 nm⁻¹ corresponds to the (001) plane of the COF TpTG_{Cl}. The peak with a q_z of 2.8 nm⁻¹ corresponds to the (100) plane of the COF TpTG_{Cl}. The (001) peak is observed as a diffuse arc of scattering concentrated near $q_{xy}=0$ and a large out-of-plane diffraction angle. The (100) peak is also observed as a diffuse arc of scattering. The result cannot provide enough evidence for the horizontal orientation of the membrane. Since the GIWAXS is principally used for characterizing the in-situ grown COF films, the top-down preparation method in our study could influence the reflections in GIWAXS pattern. We consider that the exfoliation of COFs leads to the random displacement of the 2D layers, which may hinder the pore accessibility. Furthermore, the CNFs in the membranes could also influence the (100) plane reflection of COFs in GIWAXS. Therefore the (100) peak does not exhibit centered c-axis orientations.

Figure R4. The GIWAXS pattern of the TpTG_{Cl}@CNFs-X membrane.

According to the reviewer’s comment and the GIWAXS result, we would like to revise the original description as **“The PXRD pattern of the TpTG_{Cl}@CNFs-X membrane is shown in Supplementary Fig. 17, which also displays high intensity of (100) and (001) peaks of COF TpTG_{Cl}.”** (Page 8, line 12-14).

11. The authors should explain the term “Donnan exclusion”.

Response: The Donnan exclusion is a terminology for the behavior of charged particles near a semi-permeable membrane that sometimes fail to distribute evenly across the two sides of the membrane. In desalination, the Donnan exclusion is usually caused by the charged groups fixed on the membrane to exclude the ions with the same charge. The explanation for “Donnan exclusion” has been added in the revised manuscript: **“The rejection mechanism could be explained by a synergistic effect of size sieving and Donnan exclusion caused by the charged groups on the membrane surface to exclude the ions with the same charge.”** (Page 18, line 13-15).

12. The authors mention in the “Methods” section that several other COF-CNF composite membranes were prepared from a variety of COFs, but the acronyms used to describe these materials are never defined and no characterization data is included for these alternative structures outside of the supporting information. These should be discussed in the manuscript.

Response: Additional characterization data including AFM, FTIR, NMR of COFs, and SEM images and mechanical properties of these COFs@CNFs membranes have been added in the revised manuscript. Limited by the number of figures in the manuscript, these results have been placed in the supporting information as shown in **Supplementary Figure 38-41 and**

Supplementary Table 5. The acronyms of other COFs@CNFs membranes have been defined and the characterization data for these membranes have been discussed in the manuscript (**Page 19, first paragraph in Discussion**) as shown below.

We described the fabrication of COF membranes through a mixed-dimensional assembly of 2D COF nanosheets and 1D CNFs. This unique design resulted in the fast and highly selective water transport as well as the high stability of the membranes. This assembly strategy can also be applied to other 2D COFs to fabricate robust COF-based membranes. To further validate this novel idea, another four kinds of Schiff-base type COFs i.e. TpPa-1, TpBD, TpHZ and TpBD(OH)₂⁵⁰⁻⁵² were synthesized by Tp and p-phenylenediamine (Pa-1), benzidine (BD), hydrazine hydrate (HZ), or 3,3'-dihydroxybenzidine (BD(OH)₂). The molecular structures and morphology of these COFs are shown in Supplementary Fig. 38. The structures of these COFs were confirmed by PXRD, ¹³C CP-MAS solid-state NMR and FTIR as shown in Supplementary Figs. 39 and 40. Using the same method, these COF nanosheets were also assembled with CNFs to form stable COFs@CNFs nanocomposites. Additional four kinds of COF membranes including TpPa-1@CNFs/PAN, TpBD@CNFs/PAN, TpHZ@CNFs/PAN and TpBD(OH)₂@CNFs/PAN were fabricated and these membranes showed dense and defect-free structures (Supplementary Fig. 41). These membranes also display significantly improved mechanical strength compared with the corresponding pristine COF membrane as shown in Supplementary Table 5. Furthermore, these membranes could also exhibit high permeance as well as high rejection for dyes and salts, as shown in Supplementary Table 6.

Supplementary Figure 38. Molecular structures, AFM images and height profiles of (a, a1, a2) COF TpPa-1, (b, b1, b2) TpBD, (c, c1, c2) TpHZ and (d, d1, d2) TpBD(OH)₂.

Supplementary Figure 39. PXRD patterns of the exfoliated COF TpPa-1, TpBD, TpHZ and TpBD(OH)₂.

Supplementary Figure 40. a) ¹³C CP-MAS solid-state NMR spectrum and b) FTIR spectrum of COF TpHZ, TpBD, TpPa-1 and TpBD(OH)₂.

Supplementary Figure 41. Tyndall scattering effect in the aqueous solution of (a) TpPa-1@CNFs, (b) TpBD@CNFs, (c) TpHZ@CNFs and (d) TpBD(OH)₂@CNFs nanocomposites; The digital photos, cross-sectional SEM images and surface SEM images of (a1, a2, a3) TpPa-1@CNFs/PAN membrane, (b1, b2, b3) TpBD@CNFs/PAN membrane, (c1, c2, c3) TpHZ@CNFs/PAN membrane and (d1, d2, d3) TpBD(OH)₂@CNFs/PAN membrane.

Supplementary Table 5. Mechanical properties of another four kinds of freestanding COFs@CNFs membranes and the pristine freestanding COF membranes.

Membrane	Tensile strength (MPa)	Young's modulus (GPa)
TpPa-1@CNFs-5	112.3±5.1	4.2±0.1
TpBD@CNFs-5	95.6±4.4	3.8±0.2
TpHZ@CNFs-5	107.5±6.8	4.2±0.1
TpBD(OH) ₂ @CNFs-5	98.2±6.1	4.0±0.2
TpPa-1	20.8±0.7	1.8±0.1
TpBD	21.2±0.5	2.0±0.1
TpHZ	14.6±0.3	1.7±0.1
TpBD(OH) ₂	25.3±0.4	2.0±0.2

13. The authors should rationalize why some TpTG_{Cl}@CNFs-X membranes perform better in certain applications (solvent dehydration, dye rejection, and desalination) than others with more or less CNFs.

Response: For solvent dehydration, the separation performance of the TpTG_{Cl}@CNFs-X/PAN membranes with different CNF contents is shown in Supplementary Figure 23. With the increase in CNF content, the separation factor continuously increases, while the permeation flux first increases and then decreases. The molecular transport across membrane is based on the solution-diffusion and molecular sieving mechanisms. On one hand, the incorporation of hydrophilic CNFs could enhance the solubility selectivity for water and reduce the resistance for water permeation. On the other hand, the interlamellar micropore size in membranes decreases with increasing CNF contents. Therefore, compared with the TpTG_{Cl}@CNFs-5 membrane with the optimized performance, the membranes with fewer CNFs exhibit lower separation factor, and those with more CNFs exhibit lower permeation flux.

Supplementary Figure 23. a) Ethanol dehydration (at 76 °C), b) isopropanol dehydration (at 80 °C), and c) n-butanol dehydration (at 80 °C) performance of the TpTGCl@CNFs-X/PAN membranes.

For dye rejection, the separation performance of the TpTGCl@CNFs-X/PAN membranes with different CNF contents is shown in Supplementary Figure 32. With the increase of CNF contents, the permeance continuously decreases, while the dye rejection first increases and then keeps steady. The mechanism for rejecting dyes is mainly based on molecular sieving. The interlamellar equivalent micropore size of the TpTGCl@CNFs-3 membrane is 0.82 nm, which can easily reject the dyes with sizes larger than 1 nm. The membranes with fewer CNFs have lower rejection for dyes due to the larger interlamellar pore size. Although the membranes with more CNFs also have high rejection for dyes, their permeance is much low due to the densely packed CNF networks.

Supplementary Figure 32. The dye rejection performance of the TpTG_{Cl}@CNFs-X/PAN membranes.

For desalination, the separation performance of the TpTG_{Cl}@CNFs-X membranes with different CNF contents is shown in Supplementary Figure 34. The rejection mechanism is based on a synergistic effect of size sieving and Donnan exclusion. With the increase of CNF contents, the amounts of negatively charged groups on membrane surface increase meanwhile the interlamellar micropore size decreases. Therefore, the rejection for salts increases while the permeance decreases with the increase of CNF contents.

Supplementary Figure 34. The desalination performance of the TpTG_{Cl}@CNFs-X/PAN membranes.

According to the reviewer's valuable suggestion, the above results and discussion have been added in the revised manuscript (**Page 12, line 13-18; Page 17, line 2-5; Page 18, line 21-23**).

14. The authors should more thoroughly explain why Na₂SO₄ is rejected by the TpTG_{Cl}@CNFs-4 membrane but the chloride salts are not.

Response: The salt rejection mechanism can be explained by a synergistic effect of Donnan exclusion and size sieving. The residual carboxyl groups of CNFs on the membrane surface make the surface negatively charged. These negatively charged groups on the membrane could cause an electrostatic attraction with the counter-ions, which gives rise to a concentration difference of ions across the membrane and in the solution, thus a potential difference is generated. This potential difference further repels the co-ions from the membrane and rejects the counter-ions. Therefore, the Donnan exclusion effect for sulfate salts is more notable than that for chloride salts. Furthermore, the hydration radius of Cl⁻ (0.66 nm) is smaller than that of SO₄²⁻ (0.76 nm). Driven by this Donnan exclusion combined with steric hindrance, a high rejection for sulfate salts and a relatively low rejection for chloride salts are obtained. The above explanation has been added in the revised manuscript (**Page 18, line 18-21**).

Reviewer #2

The development of covalent organic framework (COF) membranes for molecular separation has become a hot topic in membrane science and technology in recent years. This paper reports COFs@CNFs composites as a membrane material for possible applications in butanol pervaporation dehydration as well as dye separation and desalination. The characterization of the COFs@CNFs composite material is excellent, and the composite membranes indeed show improved performance in comparison with COF membranes. However, the concept for preparation of the composite membranes with addition of CNFs is quite common, which has been widely reported in the literatures for various membrane materials. Therefore, taking this point into consideration, the novelty of this paper remains an important concern for publication in this journal.

Response: Thanks for the reviewer's valuable guidance. We are sorry for the insufficient statement for the novelty in the original manuscript.

The development of 2D COF membranes with high performance as well as high stability is highly innovative in separation and other related fields. To date, only three works about integrating 2D COF nanosheets into membranes have been reported (W. X. Zhang et al, *J. Mater. Chem. A*, 2018, 6, 13331-13339; G. Li et al, *ACS Appl. Mater. Interfaces*, 2017, 9, 8433-8436; Y. P. Ying et al, *J. Mater. Chem. A*, 2016, 4, 13444-13449.) Developing an effective approach to simultaneously implement the effective pore size adjustment and mechanical strength enhancement of 2D COF membranes is still a challenge. Also, the 2D COF membranes used for liquid mixture separation and ion separation have rarely been reported.

The emergence of mixed-dimensional heterostructures (2D + n D, where n is 0, 1 or 3) has opened new avenues for fundamental scientific studies and applied device designs (D. Jariwala et al, *Nat. Mater.*, 2017, 16, 170-181). Considering the 1D materials can reduce the size of pore entrance of 2D COF nanosheets by the sheltering effect, in our work, we proposed for the first time to use 1D CNFs combined with 2D COF nanosheets to construct mixed-dimensional nanocomposites. These mixed-dimensional nanocomposites were then assembled into membranes with tunable interlamellar micropore size and improved mechanical strength. As for the reviewer's concern on the concept of preparation of composite membranes with addition of CNFs, we have carefully checked relative literatures (for example, M. Matsumoto et al, *Adv. Mater.*, 2016, 28, 1765-1769; X. F. Zhang et al, *J. Membr. Sci.*, 2018, 568, 10-16; C. Zhu et al, *ACS Appl. Mater. Interfaces*, 2017, 9, 21048-21058.). In these literatures, the CNFs were mainly used as matrices to fabricate composite membranes, and the entangled web-like structures of CNFs were effective to reduce the voids in the membranes. Therefore, both the preparation methods of membranes and the roles of CNFs in our work are different from those reported

composite membranes with addition of CNFs. Furthermore, the membranes in our work show superior n-butanol dehydration and dye or sulfate salt rejection performance. We believe such an improvement is significant in the field of 2D COF membranes.

1. Compared with the water contact angles of PAN, TpTG_{Cl}/PAN and CNFs/PAN, why the water contact angle of TpTG_{Cl}@CNFs-5/PAN is much smaller? The authors should clearly explain the relationship between these properties and the material structures, which would be helpful for understanding water transport mechanism of the composite membrane.

Response: The water contact angles of these four kinds of membranes are shown in Supplementary Figure 25.

Supplementary Figure 25. Water contact angles of the TpTG_{Cl}@CNFs-5/PAN, PAN, pristine TpTG_{Cl}/PAN and pristine CNFs/PAN membranes within 0.2 s.

The PAN substrate shows the largest water contact angle of 71° mainly due to the low hydrophilicity of the cyano groups.

The pristine TpTG_{Cl}/PAN membrane shows a low average surface roughness (R_a) of 14.8 as shown in Figure R5. When a droplet of water was brought in contact with the surface of the membrane in air, the advancing water contact angle appeared to be 29°. The water droplet

quickly spread and permeated into the membrane within 0.2 s. The ultrafast water permeation can be explained by the capillary effect of the membrane according to the Laplace theory:

$$P_c = \Delta p = -2\gamma \cos \theta_\alpha / R \quad (\text{Equation R1})$$

Where Δp is the liquid intrusion pressure, γ is the surface tension of water in air, θ_α is the advancing contact angle of the water on the surface, R is the equivalent pore radius of the membrane. Since the θ_α is $< 90^\circ$, the calculated P_c is < 0 . Meanwhile, the small pore size (~ 1.3 nm) of TpTG_{Cl}/PAN membrane results in a large absolute value of P_c , which means that water is easy to permeate into the membrane with the capillary driving pressure of P_c . The above analyses can well explain the fast water spreading ability.

Membrane	Roughness	
	R_a (nm)	R_q (nm)
TpTG _{Cl} /PAN	14.8±0.8	18.3±1.1
CNFs/PAN	66.3±1.8	50.6±2.0
TpTG _{Cl} @CNFs-5/PAN	69.4±3.7	56.3±2.9

Figure R5. 3D AFM images of the a) TpTG_{Cl}/PAN, b) CNFs/PAN and c) TpTG_{Cl}@CNFs/PAN-5 membrane with a scan area of $5 \mu\text{m} \times 5 \mu\text{m}$, and the corresponding average roughness (R_a) and root-mean-square roughness (R_q).

The pristine CNFs/PAN membrane also demonstrated a hydrophilic surface with an advancing water contact angle of 31° , which can be attributed to the high water adsorption capacity of CNFs ($675 \text{ cm}^3 \text{ g}^{-1}$ at $P/P_0=0.9$) as shown in Supplementary Figure 27 and the rough surface (R_a of 66.3). However, the fast water spreading phenomenon was not observed for CNFs/PAN membrane which could be because the water permeation in CNFs was much slower than that in TpTG_{Cl}.

Supplementary Figure 27. Water vapor adsorption-desorption isotherms of TpTG_{Cl} and CNFs measured at 298 K.

The TpTG_{Cl}@CNFs-5/PAN membrane shows a much smaller water contact angle (7.2°) than the CNFs/PAN membrane (31°) and TpTG_{Cl}/PAN membrane (29°). There is no significant difference in surface roughness between the CNFs/PAN membrane and TpTG_{Cl}@CNFs-5/PAN membrane as shown in Figure R5. Therefore, the small water contact angle for the TpTG_{Cl}@CNFs-5/PAN membrane could be due to the synergy between TpTG_{Cl} and CNFs. The TpTG_{Cl}@CNFs-5/PAN membrane integrates with both high water adsorption of CNFs and fast water spreading ability of TpTG_{Cl}, thus showing the smallest water contact angle.

The relationship between the water contact angle and these material structures has been explained in the revised manuscript (**Page 14, line 6-Page 15, line 10**).

2. In addition to pore size distribution, what is average pore size of the membranes? Which is very important to understand the transport behaviors of the membranes.

Response: The average pore sizes of the membranes were calculated by NL-DFT methods from the N₂ adsorption-desorption isotherms as shown in Table R1.

Table R1 The average pore size of the membranes.

Membrane	Average pore size (nm)
TpTG _{Cl} @CNFs-1	1.35
TpTG _{Cl} @CNFs-2	1.32
TpTG _{Cl} @CNFs-3	1.26
TpTG _{Cl} @CNFs-4	1.24
TpTG _{Cl} @CNFs-5	1.21

The average pore size is a statistical average of pore sizes in membrane, while the pore size distribution displays the relative number of pores with different sizes (Figure 3b and Supplementary Figure 18). We consider the pore size distribution of the membranes could more comprehensively elucidate the transport behaviors of the membranes compared with the average pore size results. Therefore, the pore size distribution results of the membranes are adopted in this work. For all membranes, the pore size distribution of about 1.3 nm is derived from the intrinsic pores of TpTG_{Cl}. The pore size distribution of less than 1.3 nm is derived from the interlamellar micropores in membranes. The interlamellar equivalent micropore size of the membranes has been effectively manipulated in the range of 0.45-1.0 nm by the sheltering effect of 1D cellulose nanofibers (CNFs). The large intrinsic pores of TpTG_{Cl} provided fast molecular transfer channels, while the small interlamellar micropores afford molecular sieving ability.

3. The long-term stability of TpTG_{Cl}@CNFs-5/PAN membrane seems a big concern, why the water permeation flux continuously decreases with operating time for butanol dehydration?

Response: The initial water concentration in feed is 10 wt%. During the long-term operation, the water in feed continuously transported across the membrane, thus the water concentration in feed gradually decreased. The decreased water concentration in feed resulted in reduction of the partial pressure of water at the upstream side. Therefore, a 10% decline in permeation flux was observed after 15 d as shown in the original Fig. 3b. Similar results also have been reported in the literature (J. J. Yang et al., *Adv. Mater.*, 2018, 1705775; K. C. Guan et al., *ACS Appl. Mater. Interface*, 2018, 10, 13903-13913).

Original Fig. 3b. Long-term permeation flux and water concentration in permeate of the TpTG_{Cl}@CNFs-5/PAN membrane applied in a water/n-butanol system (water concentration in feed: 10 wt%; operating temperature: 80 °C).

To eliminate the influence of partial pressure driving force, the permeance ((P/l) , GPU) of the membrane was calculated by Equation (4) and the results are shown in Figure R6.

$$(P/l)_i = \frac{J_i}{p_{i0} - p_{il}} = \frac{J_i}{\gamma_{i0} x_{i0} P_{i0}^{sat} - p_{il}} \quad (4)$$

where l refers to the membrane thickness (m). For individual component i , J_i is the permeation flux ($\text{kg m}^{-2} \text{h}^{-1}$), p_{i0} and p_{il} are the partial pressure (Pa) in the upstream side and the downstream side, respectively. γ_{i0} refers to the activity coefficient which is calculated by the Wilson equation. x_{i0} is the mole fraction in the feed and P_{i0}^{sat} is the saturated vapor pressure obtained through the Antoine equation.

Figure R6. Long-term permeance and water concentration in permeate of the TpTGCl@CNFs-5/PAN membrane applied in a water/n-butanol system (water concentration in feed: 10 wt%; operating temperature: 80 °C).

From Figure R6, it can be seen that both the permeance and the water concentration in permeate are very stable during 15 d, demonstrating the high long-term stability of the membrane. **The original Fig. 3b has been replaced by the Figure R6.**

4. There is no information about long-term stability of dye separation and desalination performance. It is better to add additional experiments to confirm the stability in above systems, since the membrane is not so stable for butanol dehydration.

Response: The long-term stability of dye rejection and desalination was evaluated as shown in Figure R7 and R8. The nanofiltration experiments were carried out on a small vacuum filtration device in which the feed with constant concentration was continuously provided. No significant loss in the permeance was observed over 150 h and the rejection for congo red and Na_2SO_4 remained constant.

Figure R7. Long-term stability of the TpTGCl@CNFs-3/PAN membrane for rejecting congo red (dye concentration: 100 ppm; applied pressure: 0.2 MPa; room temperature).

Figure R8. Long-term stability of the TpTGCl@CNFs-4/PAN membrane for rejecting Na₂SO₄ (salt concentration: 1000 ppm; applied pressure: 0.4 MPa; room temperature).

According to the reviewer's valuable suggestion, these results have been added into the supplementary information (**Supplementary Figure 33 and 35**) and the revised manuscript (**Page 17, line 5-7; Page 19, line 3-4**).

5. Regarding the membrane desalination performance, although the COFs@CNFs composite membranes show a high permeation flux and acceptable Na₂SO₄ rejection, but the rejection for CaCl₂ and NaCl is quite low. As a matter of fact, so many membranes have been reported for desalination with both high CaCl₂ and NaCl rejection. The comparison of desalination

performance should involve the CaCl₂ and NaCl systems in order to objectively judge the COFs@CNFs composite membrane performance for desalination application.

Response: The comparison of CaCl₂ and NaCl rejection performance of some representative membranes reported in the literature and in this study is shown in Table R2 and R3.

Table R2 The comparison of CaCl₂ rejection performance of representative membranes reported in the literature and in this study.

Membrane type	Permeance (L m ⁻² h ⁻¹ bar ⁻¹)	Rejection for CaCl ₂ (%)	Reference
PA membrane	36	62.8	Small, 2016, 36, 5034-5041
PA membrane	17.6	60	J. Membr. Sci., 2016, 515, 238-244
GO-based membrane	12.5	98.3	ACS Nano, 2018, 9, 9309-9317
PA membrane	19.8	92.7	Science, 2018, 360, 518-521
PA membrane	52	28	Nat. Commun., 2018, 9, 1-9
PA membrane	11.4	32.9	J. Mater. Chem. A, 2018, 6, 6442-6454
COF-based membrane	41.6	40.3	This work

Table R3 The comparison of NaCl rejection performance of representative membranes reported in the literature and in this study.

Membrane type	Permeance (L m ⁻² h ⁻¹ bar ⁻¹)	Rejection for NaCl (%)	Reference
GO-based membrane	9.51	51.4	ACS Appl. Mater. Interfaces, 2017, 9, 41482-41495
PA membrane	25	53	J. Membr. Sci., 2017, 524, 174-185
PA membrane	2.2	60	J. Membr. Sci., 2009, 326, 19-26
GO-based membrane	12.5	97	ACS Nano, 2018, 12, 9309-9317
PA membrane	25.8	49.6	Science, 2018, 360, 518-521
CNT TFN membrane	13.2	45	J. Membr. Sci., 2017, 524, 344-353
PA membrane	23.5	37.1	J. Membr. Sci., 2016, 502, 106-115

PA membrane	14	40	J. Membr. Sci., 2017, 523, 282-290
PA membrane	12.1	30	J. Membr. Sci., 2016, 498, 374-384
PA membrane	54	11	Nat. Commun., 2018, 9, 1-9
COF-based membrane	42.2	24.2	This work

We would like to emphasize that the nanofiltration membrane is mostly used for monovalent/divalent ion separations, which is different from the reverse osmosis membrane that rejects all ions. Na₂SO₄ and MgSO₄ are the two common salts being used to evaluate the desalination performance of nanofiltration membranes. (B. S. Hsiao, et al., J. Membr. Sci. 2009, 326, 484-492; S. B. Zhang, et al., J. Membr. Sci. 2009, 335, 133-139; J. N. Shen, et al., J. Membr. Sci. 2013, 442, 18-26; X. L. Wang, et al., ACS Nano 2015, 9, 7488-7496; Z. K. Xu, et al., J. Mater Chem. A 2017, 5, 16289-16295), even for commercial nanofiltration membranes, such as Dow® and Nitto®. Because most of Nanofiltration membranes including polyamide (PA) are negatively charged, they usually exhibit high rejection to divalent anions driven by the size sieving and Donnan exclusion caused by the charged groups fixed on the surface to exclude the ions with same charge. Meanwhile, when a nanofiltration membrane shows a high rejection towards sulfate salts and a low rejection to chloride salts simultaneously, it is an important parameter for NF membrane and is advantageous in practical application due to the high ion selectivity (M. L. Bruening, et al., J. Membr. Sci. 2006, 283, 366-372; M. L. Bruening, et al., Ind. Eng. Chem. Res. 2006, 45, 6284-6288).

The TpTG_{Cl}@CNFs-4/PAN membrane in our study displays high rejection for Na₂SO₄ (96.8%) and for MgSO₄ (93.2%), and relatively low rejection for CaCl₂ and NaCl. The rejection for NaCl is quite low (only 24.2%) but the permeance is higher than 40 L m⁻² h⁻¹ bar⁻¹. In comparison with the rejection behavior for Na₂SO₄ and MgSO₄, the as-prepared TpTG_{Cl}@CNFs-4/PAN membrane shows a high ion selectivity to monovalent ions and divalent ions, demonstrating their potential application in desalination of solutions containing high concentration multivalent salts.

Reviewer #3

Overall, this paper is in its current state quite confusing. Many (crucial) aspects are unclear. Performance nor preparation method seem at this moment good enough to me to make serious impact.

Response: Thanks this reviewer for the valuable guidance for our manuscript. We apologize for the unclear statements in the original manuscript. We have made the effort to improve the clarity in the revised manuscript and have also included essential control experiments and characterization data. The novelty of our study has been further strengthened based on the more systematic experimental results as well as the more in-depth analysis. We would like to emphasize the novelty of our work again.

The development of 2D COF membranes with high performance as well as high stability is highly innovative in separation and other related fields. To the best of our knowledge, the 2D COF membranes used for liquid mixture separation and ion separation have rarely been reported. Developing an effective approach to simultaneously implement the effective pore size adjustment and mechanical strength enhancement of 2D COF membranes is imperative. The emergence of mixed-dimensional heterostructures ($2D + nD$, where n is 0, 1 or 3) has opened new avenues for fundamental scientific studies and applied device designs (D. Jariwala et al, Nat. Mater., 2017, 16, 170-181). Considering the 1D materials can reduce the size of pore entrance of 2D COF nanosheets by the sheltering effect, in our work, we proposed for the first time to use 1D CNFs combined with 2D COF nanosheets to construct mixed-dimensional nanocomposites. These mixed-dimensional nanocomposites were then assembled into membranes with tunable interlamellar micropore size and improved mechanical strength. The as-prepared membranes exhibit high separation performance in various molecular separations, including alcohol dehydration, removal of organic dyes and desalination. In particular, the TpTG_{Cl}@CNFs-5/PAN membrane displays a high permeation flux of $8.53 \text{ kg m}^{-2} \text{ h}^{-1}$ with a separation factor of 3876 for n-butanol dehydration, which is superior to most reported membranes. The TpTG_{Cl}@CNFs-4/PAN membrane exhibits the high permeance of $42.8 \text{ L m}^{-2} \text{ h}^{-1} \text{ bar}^{-1}$ with a rejection of 96.8% for Na_2SO_4 at a low pressure, which is also superior to most reported nanofiltration membranes. Our design strategy could extend the application scope of COF membranes in energy and environmental fields. We do think such an improvement is significant in the field of 2D COF membranes.

1. I find the link with 'bio-inspired' a bit far-fetched. I understand it is a popular selling point, but here I find it a bit exaggerated for use. The honeycomb is just a direct consequence of the chosen building blocks, no?

Response: We agree to the reviewer's opinion and we have removed the original statements about "honeycomb-inspired architecture".

2. Is the Tyndall effect relevant enough to mention here so explicitly?

Response: The original description "The as-prepared TpTG_{Cl}@CNFs-X colloidal suspension displays a clear Tyndall scattering effect (Fig. 1g)." has been revised to "**The TpTG_{Cl}@CNFs-X nanocomposites can be easily dispersed in water to form a stable colloidal solution (Fig. 1g), which is a preferential precursor for membrane fabrication.**" (Page 6, line 1-3).

3. Is this really self-assembly? It is only the COF that self-assembles, right? Not the full membrane?

Response: Yes, as shown in the following Fig. 2a, the mixed-dimensional nanocomposites (COFs@CNFs-X) were prepared by the self-assembly, and the membranes were also fabricated by the self-assembly.

Fig. 2a A schematic illustration showing the vacuum-assisted self-assembly and the mixed-dimensional nanostructure.

The COF nanosheets were initially self-assembled with CNFs to form stable mixed-dimensional nanocomposites, which consisted of a COF nanosheet uniformly encapsulated into a dense network of 1D CNFs.

The COFs@CNFs-X nanocomposites were used for fabricating membranes by a vacuum-assisted self-assembly method as shown in Fig. 2a. In this procedure, the COFs@CNFs-X nanocomposites were stacked into the membrane in one-step. Therefore, both the mixed-dimensional nanocomposites and the membranes were formed by the self-assembly.

4. Better schemes are required to explain the transport. Is it only the interlamellar micropores that are relevant for the transport? Not the ring in the COF?

Response: The transport mechanism scheme is shown in Figure R9, both the interlamellar micropores and the intrinsic channels of COF (ring in the COF) play important roles. The interlamellar micropores endow the membranes with molecular sieving properties, while the intrinsic channels of COF provide rapid transport pathways for water. **The original scheme in Fig. 4d has been modified.**

Figure. R9. A schematic diagram of liquid molecules transport through the membrane.

5. SuppTable 4: Quite sure better PA-membranes have been reported in terms of selectivity for sure. Pressures are all so low, far below real desalination!!

Response: The data in Supplementary Table 4 were collected based on the representative nanofiltration membranes for Na_2SO_4 rejection. In light of the reviewer's comment, some recently reported PA-membranes with higher rejection have also been listed in the revised Supplementary Table 4. Compared with these reported membranes, the Na_2SO_4 rejection of our membranes is acceptable, and the permeance is also higher than those of most reported membranes.

We would like to emphasize that the nanofiltration membranes are mostly used for monovalent/divalent ion separations at a relative low pressure. The development of low-energy consumption and environment-friendly membrane techniques for the desalination of wastewater

and seawater is critical. Nanofiltration membranes have served as an excellent alternative due to their low pressure and high flux for monovalent/divalent ion separations.

6. I could not find how the selectivity was calculated in PV.

Response: The permeance and selectivity in PV were calculated by equation (4) and (5) (Page 23, line 20-Page 24, line 8).

To eliminate the influence of partial pressure driving force, the permeance ((P/l) , GPU) of the individual component were calculated by Equation (4) and selectivity (β) is measured by Equation (5).

$$(P/l)_i = \frac{J_i}{p_{i0} - p_{il}} = \frac{J_i}{\gamma_{i0} x_{i0} p_{i0}^{sat} - p_{il}} \quad (4)$$

$$\beta = \frac{(P/l)_W}{(P/l)_A} \quad (5)$$

where l refers to the membrane thickness (m). For individual component i , J_i is the permeation flux ($\text{kg m}^{-2} \text{h}^{-1}$), p_{i0} and p_{il} are the partial pressure (Pa) in the upstream side and the downstream side, respectively. γ_{i0} refers to the activity coefficient which is calculated by the Wilson equation. x_{i0} is the mole fraction in the feed and p_{i0}^{sat} is the saturated vapor pressure obtained through the Antoine equation. Subscript W and A stand for water and alcohol, respectively.

7. The role of Donnan exclusion: it is explained like ions are attracted to the membrane. This is undesired, no?

Response: The Donnan exclusion refers to the behavior of charged particles near a semi-permeable membrane that sometimes fail to distribute evenly across the two sides of the membrane. In desalination, the Donnan exclusion is usually caused by the charged groups fixed on the membrane to exclude the ions with the same charge. The residual carboxyl groups of CNFs on the membrane surface make the surface negatively charged. These negatively charged groups on the membrane could cause an electrostatic attraction with the counter-ions, which gives rise to a concentration difference of ions across the membrane and in the solution, thus a potential difference is generated. This potential difference further repels the co-ions from the membrane and rejects the counter-ions. Salts rejection is thus achieved.

8. In addition, language is often not 100% correct or clear.

E.g. First line second paragraph in abstract (I guess it is referring to porous materials only?)

Response: If our guessing is correct, the reviewer may refer to the first sentence “Covalent organic frameworks (COFs) with rigid skeleton and engineered pore hold great promise in molecular separation of liquid mixtures.”. Accordingly, this sentence has been revised to “**Covalent organic frameworks (COFs) hold great promise in molecular separations owing to their robust, ordered and tunable porous network structures.**” (Page 2, line 1-2)

Line 43: 'greater than': shouldn't this be the case for permeance?

Response: The phrase “greater than” has been revised to “**larger than**”. (Page 3, line 8)

64: what is meant with a 'sheltering effect'?

Response: The CNFs covered onto the surface of COF nanosheets which sheltered the pores of COF nanosheets caused the decreased pore size of COF nanosheets as shown in Figure R10. We referred to this phenomenon as "sheltering effect".

Figure R10. A schematic illustration of the “sheltering effect”.

9. Minor:

Refs sometimes contain 'et al.'

Response: The formats of references we used are according to the standard Nature referencing style: **All authors should be included in reference lists unless there are six or more, in which case only the first author should be given, followed by 'et al.'**

10. Many figures are not clear, especially when there is text in them, which can not be well read.

Response: According to the reviewer’s valuable suggestion, we have carefully checked and revised all of the unclear figures.

The figures such as **Fig. 1d-h, 2a, 4d, Supplementary Fig. 5, 8f, 11, 20, 23, and 24** have all been revised.

Reviewers' comments:

Reviewer #2 (Remarks to the Author):

Although the authors have made significant modifications in the revised manuscript, important concerns regarding the characterization and permeation mechanism of still remain.

1. The water contact angles of the membranes are still very much confusing. Judging from the color of the samples of TpTGCl@CNF-5/PAN and TpTGCl/PAN membranes in Figure S25, the coverage of the TpTGCl/PAN membrane is obviously much poorer than that of the TpTGCl@CNF-5/PAN membrane, and the PAN support seems to be not completely covered by TpTGCl. The higher water contact angle of TpTGCl@CNF-5/PAN is possibly ascribed to the exposure of the support, because the support shows a high water contact angle. Therefore, the water contact angle of TpTGCl@CNF-5/PAN membrane and related discussion is thus not reliable in this paper. The authors should prepare TpTGCl@CNF-5/PAN and TpTGCl/PAN membranes with full coverage for characterization and discussion.

2. The proposed permeation mechanism and experimental data are contradictory. The authors claimed that the water molecules are first preferentially captured by CNFs, and then permeate through the channels of TpTGCl in through-plane direction. However, based on the water vapor adsorption-desorption isotherms in Figure S27, the water adsorption amount is higher for TpTGCl at a relative low pressure, and water molecules should be preferentially adsorbed on TpTGCl because water concentration in the feed is only 10wt%. Furthermore, the authors also claimed that the CNFs in the interlayer concentrate the water molecules to further enhance the solubility selectivity, which is also not consistent with the higher adsorption amount on TpTGCl than that of CNFs at a low relative pressure.

Reviewer #3 (Remarks to the Author):

The authors surely did a lot of effort to get their paper improved. By the way, I would strongly encourage them to be shorter in the responses next time and simply refer to the earlier place where aspects were explained for another reviewer (e.g. Donnan exclusion is now discussed 2x).

With especially comments on novelty and performance coming back from more than 1 reviewer, I would not have expected this paper to be allowed to pass. These aspects remain the same. They are maybe a bit better framed, but if statements like 'rarely reported', 'superior to most reported membranes', 'an improvement' have to be used, I would have anticipated that more novelty and better performances would have been required to enter this prestigious journal.

But the authors surely did clarify some points and tried to improve figures, so I assume it can be accepted now.

Maybe still replace 'desalination' by 'salt rejection' or 'nanofiltration'. NaCl rejection is really not good (even the Na₂SO₄ rejection is not special) and that is what you expect from 'desalination'.

[Note from the Editor: Reviewer #3 was asked to look over the answers given to reviewer #1]

Reviewer #2 (Remarks to the Author):

Although the authors have made significant modifications in the revised manuscript, important concerns regarding the characterization and permeation mechanism still remain.

1. The water contact angles of the membranes are still very much confusing. Judging from the color of the samples of TpTG_{Cl}@CNF-5/PAN and TpTG_{Cl}/PAN membranes in Figure S25, the coverage of the TpTG_{Cl}/PAN membrane is obviously much poorer than that of the TpTG_{Cl}@CNF-5/PAN membrane, and the PAN support seems to be not completely covered by TpTG_{Cl}. The higher water contact angle of TpTG_{Cl}/PAN is possibly ascribed to the exposure of the support, because the support shows a high water contact angle. Therefore, the water contact angle of TpTG_{Cl}/PAN membrane and related discussion is thus not reliable in this paper. The authors should prepare TpTG_{Cl}@CNF-5/PAN and TpTG_{Cl}/PAN membranes with full coverage for characterization and discussion.

Response: Thank the reviewer very much for the valuable guidance. We have acquired the surface and cross-sectional SEM images of the TpTG_{Cl}/PAN membrane (corresponding to Figure S25), as shown in Figure R1. It can be seen that the PAN support is completely covered by TpTG_{Cl} for the TpTG_{Cl}/PAN membrane, which demonstrates a void-free or defect-free surface morphology (Figure R1a) and a continuous TpTG_{Cl} layer with a uniform thickness of around 200 nm on the PAN support (Figure R1b and c).

Figure R1. (a) Surface and (b-c) cross-sectional SEM images of the TpTG_{Cl}/PAN membrane.

To further prove the reliability of the water contact angle results, three replicate TpTG_{Cl}/PAN membranes with a higher amount of TpTG_{Cl} (0.42 mg) were fabricated and their water contact angles were measured. The digital photos of these membranes are shown in Figure R2. If the coverage is incomplete for the previous TpTG_{Cl}/PAN membrane, there would be a significant difference in the water contact angle between the previous and replicate TpTG_{Cl}/PAN membranes. For each replicate TpTG_{Cl}/PAN membrane, the water contact angle was measured 5 times and the mean values and standard deviation are given in Figure R3. It can be seen that the water contact angles (29.3° at T=0.05 s and 10.4° at T=0.2 s) of the replicate TpTG_{Cl}/PAN membrane with a higher amount of TpTG_{Cl} (0.42 mg) are very similar to that of the previous TpTG_{Cl}/PAN membrane with a TpTG_{Cl} amount of 0.14 mg (29.0° at T=0.05 s and 9.8° at T=0.2 s), thereby verifying the complete coverage of TpTG_{Cl} for the previous TpTG_{Cl}/PAN membrane.

Figure R2. The digital photos of three replicate TpTG_{Cl}/PAN membranes.

Figure R3. Water contact angle of the replicate TpTG_{Cl}/PAN membrane.

In addition, the TpTG_{Cl}@CNF-5/PAN membrane also has full coverage, which can be proved by its surface and cross-sectional SEM images, as shown in Figure 2. We think the above results can ensure the reliability of the data and support the related discussion.

For clarity, the original images of TpTG_{Cl}/PAN membrane in Figure S25 have been replaced, and the revised Figure S25 is shown below.

Revised Supplementary Figure 25. Water contact angles of the TpTG_{Cl}@CNFs-5/PAN, PAN, pristine TpTG_{Cl}/PAN and pristine CNFs/PAN membranes within 0.2 s.

2. The proposed permeation mechanism and experimental data are contradictory. The authors claimed that the water molecules are first preferentially captured by CNFs, and then permeate through the channels of TpTG_{Cl} in through-plane direction. However, based on the water vapor adsorption-desorption isotherms in Figure S27, the water adsorption amount is higher for TpTG_{Cl} at a relative low pressure, and water molecules should be preferentially adsorbed on TpTG_{Cl} because water concentration in the feed is only 10 wt%. Furthermore, the authors also claimed that the CNFs in the interlayer concentrate the water molecules to further enhance the solubility selectivity, which is also not consistent with the higher adsorption amount on TpTG_{Cl} than that of CNFs at a low relative pressure.

Response: Thank the reviewer very much for pointing out the contradictory descriptions between the proposed permeation mechanism and experimental data. The superhydrophilic membrane surface possesses preferential adsorption ability towards the water molecules in the feed (90 wt% alcohol aqueous solution). Therefore, compared with alcohol molecules, water molecules in the feed could be preferentially adsorbed on the membrane surface. According to the water vapor adsorption-desorption isotherms shown in Figure S27, the water adsorption amount of TpTG_{Cl} is higher than that of CNFs at a low relative pressure, thus the TpTG_{Cl} on the surface could contribute more towards water adsorption than the CNFs, just as the reviewer pointed out. The feed would then rapidly transport through the

well-organized channels of TpTG_{Cl} in through-plane direction. The CNFs in the interlayer and the next TpTG_{Cl} nanosheets can concentrate the water molecules to further enhance the solubility selectivity for water. The sheltering effect of CNFs reduces the size of pore entrance of TpTG_{Cl} framework and the interlamellar micropores with molecular sieving effects enhance the diffusion selectivity for water.

Based on the reviewer's valuable guidance, we would like to modify the original permeation mechanism to make it accord with the experimental results. The original statement "The water molecules in feed could be preferentially captured by the hydrophilic CNFs, and then the feed would rapidly transport through the well-organized channels of TpTG_{Cl} in through-plane direction. The CNFs in the interlayer concentrate the water molecules to further enhance the solubility selectivity, meanwhile, the interlamellar micropores with molecular sieving effects enhance the diffusion selectivity." **has been revised to** "Compared with n-butanol molecules, water molecules in the feed could be preferentially adsorbed on the superhydrophilic membrane surface. According to the water vapor adsorption-desorption isotherms in Supplementary Fig. 27, the water adsorption amount of TpTG_{Cl} is higher than that of CNFs at a low relative pressure, thus the TpTG_{Cl} on the surface could contribute more towards water adsorption than the CNFs. The feed would then rapidly transport through the well-organized channels of TpTG_{Cl} in through-plane direction. The CNFs in the interlayer and the next TpTG_{Cl} nanosheets can concentrate the water molecules to further enhance the solubility selectivity for water. Meanwhile, the interlamellar micropores with molecular sieving effects enhance the diffusion selectivity for water."

Reviewer #3 (Remarks to the Author):

The authors surely did a lot of effort to get their paper improved. By the way, I would strongly encourage them to be shorter in the responses next time and simply refer to the earlier place where aspects were explained for another reviewer (e.g. Donnan exclusion is now discussed 2x).

Response: Thank the reviewer very much for the positive comments and valuable guidance on our manuscript. This time, we try to shorten the responses.

With especially comments on novelty and performance coming back from more than 1 reviewer, I would not have expected this paper to be allowed to pass. These aspects remain the same. They are maybe a bit better framed, but if statements like 'rarely reported', 'superior to most reported membranes', 'an improvement' have to be used, I would have anticipated that more novelty and better performances would have been required to enter this prestigious journal. But the authors surely did clarify some points and tried to improve figures, so I assume it can be accepted now.

Response: Thank the reviewer very much for the positive remarks, kind encouragement and valuable guidance on our manuscript.

We have carefully checked the published papers concerning COF membranes for liquid separations^{R1-R10} and found that these COF membranes are all applied in nanofiltration with a molecular weight cutoff (MWCO) of 300-1400 Da (corresponding pore size of membranes in the range of 1.2~2.3 nm) since the pore size of currently available COFs is relatively large. In our work, the interlamellar equivalent micropore size of the COF membranes has been effectively manipulated in the range of 0.45-1.0 nm by the sheltering effect of 1D cellulose nanofibers. Our work should be the first attempt of COF membranes in the separation of smaller liquid molecule mixture like alcohol dehydration and salt rejection.

According to the reviewer's valuable suggestion, the statement of "the performance of TpTG_{Cl}@CNFs-4/PAN membrane is **superior** to most of the representative membranes for rejecting Na₂SO₄" in the original manuscript has been changed into "the performance of TpTG_{Cl}@CNFs-4/PAN membrane is **comparable** to most of the representative membranes for rejecting Na₂SO₄".

References

- R1. Zhang, W. X., Zhang, L. M., Zhao, H. F., Li, B. & Ma, H. P. A two-dimensional cationic covalent organic framework membrane for selective molecular sieving. *J. Mater. Chem. A* **6**, 13331-13339 (2018).
- R2. Kandambeth, S. et al. Selective molecular sieving in self-standing porous covalent-organic-framework membranes. *Adv. Mater.* **29**, 1603945 (2017).
- R3. Shi, X. S., Xiao, A. K., Zhang, C. X. & Wang, Y. Growing covalent organic frameworks on porous substrates for molecule-sieving membranes with pores tunable from ultra- to nanofiltration. *J. Membr. Sci.* **576**, 116-122 (2019).
- R4. Dey, K. et al. Selective molecular separation by Interfacially crystallized covalent organic framework thin films. *J. Am. Chem. Soc.* **139**, 13083-13091 (2017).
- R5. Matsumoto, M. et al. Lewis-acid-catalyzed interfacial polymerization of covalent

organic framework films. *Chem* **4**, 308-317 (2018).

R6. Wang, R., Shi, X. S., Xiao, A. K., Zhou, W. & Wang, Y. Interfacial polymerization of covalent organic frameworks (COFs) on polymeric substrates for molecular separations. *J. Membr. Sci.* **566**, 197-204 (2018).

R7. Hao, Q. et al. Confined synthesis of two-dimensional covalent organic framework thin films within superspreading water layer. *J. Am. Chem. Soc.* **140**, 12152-12158 (2018).

R8. Gadwal, I. et al. Synthesis of sub-10 nm two-dimensional covalent organic thin film with sharp molecular sieving nanofiltration. *ACS Appl. Mater. Interfaces* **10**, 12295-12299 (2018).

R9. Shinde, D. B. et al. Crystalline 2D covalent organic framework membranes for high-flux organic solvent nanofiltration. *J. Am. Chem. Soc.* **140**, 14342-14349 (2018).

R10. Fan, H. W., Gu, J. H., Meng, H., Knebel, A. & Caro, J. High-flux membranes based on the covalent organic framework COF-LZU1 for selective dye separation by nanofiltration. *Angew. Chem. Int. Edit.* **57**, 4083-4087 (2018).

Maybe still replace 'desalination' by 'salt rejection' or 'nanofiltration'. NaCl rejection is really not good (even the Na₂SO₄ rejection is not special) and that is what you expect from 'desalination'.

Response: According to the reviewer's valuable suggestion, the 'desalination' in the original manuscript **has been replaced** by the 'salt rejection' in the revised manuscript.

REVIEWERS' COMMENTS:

Reviewer #2 (Remarks to the Author):

The reviewer is satisfied with the revised manuscript, and it can be accepted for publication.

REVIEWERS' COMMENTS:

Reviewer #2 (Remarks to the Author):

The reviewer is satisfied with the revised manuscript, and it can be accepted for publication.

Response: Thank the reviewer very much for the positive remarks, kind encouragement and valuable guidance on our manuscript.